# Disrupting iron homeostasis can potentiate colistin activity and overcome colistin resistance mechanisms in Gram-Negative Bacteria

Kavita Gadar[1], Rubén de Dios [1], Nikol Kadeřábková [2], Thomas A. K. Prescott[3], Despoina A. I. Mavridou [2,4] & Ronan R. McCarthy [1✉]

*Acinetobacter baumannii* is a Gram-negative priority pathogen that can readily overcome antibiotic treatment through a range of intrinsic and acquired resistance mechanisms. Treatment of carbapenem-resistant *A. baumannii* largely relies on the use of colistin in cases where other treatment options have been exhausted. However, the emergence of resistance against this last-line drug has significantly increased amongst clinical strains. In this study, we identify the phytochemical kaempferol as a potentiator of colistin activity. When administered singularly, kaempferol has no effect on growth but does impact biofilm formation. Nonetheless, co-administration of kaempferol with sub-inhibitory concentrations of colistin exposes bacteria to a metabolic Achilles heel, whereby kaempferol-induced dysregulation of iron homeostasis leads to bacterial killing. We demonstrate that this effect is due to the disruption of Fenton's reaction, and therefore to a lethal build-up of toxic reactive oxygen species in the cell. Furthermore, we show that this vulnerability can be exploited to overcome both intrinsic and acquired colistin resistance in clinical strains of *A. baumannii* and *E. coli* in vitro and in the *Galleria mellonella* model of infection. Overall, our findings provide a proof-of-principle demonstration that targeting iron homeostasis is a promising strategy for enhancing the efficacy of colistin and overcoming colistin-resistant infections.

[1] Biosciences, Department of Life Sciences, College of Health, Medicine and Life Sciences, Brunel University London, Uxbridge UB8 3PH, UK. [2] Department of Molecular Biosciences, The University of Texas at Austin, Austin, TX 78712, USA. [3] Royal Botanic Gardens, Kew, Richmond, Surrey TW9 3AB, UK. [4] John Ring LaMontagne Centre for Infectious Diseases, The University of Texas at Austin, Austin, TX 78712, USA. ✉email: ronan.mccarthy@brunel.ac.uk

*A*cinetobacter baumannii is a Gram-negative bacterium that causes pneumonia, as well as wound, bloodstream and urinary tract infections in hospital and community settings[1]. The prevalence of multidrug resistance among clinical isolates of *A. baumannii* can reach as high as 70% in certain parts of the world[2]. For this reason, the World Health Organisation has classified this organism as a critical priority pathogen for which novel therapeutics are urgently needed[3]. With our antibiotic armamentarium for *A. baumannii* infections severely diminished, clinical reliance on the last-resort antibiotic colistin for the treatment of recalcitrant infections has increased, resulting in a surge of colistin resistance in clinical strains[4–6].

Colistin was first introduced in the clinic in the 1950's, however severe side effects, such as nephrotoxicity and neurotoxicity, led to its removal from therapeutic use in the 1970's. Nonetheless, the increased prevalence of MDR pathogens over the last three decades, and in particular the continuous emergence of resistance against complex β-lactam antibiotics like carbapenems, has forced clinicians to re-introduce colistin for the treatment of challenging infections[7–10]. Despite needing to resort to treating patients with colistin, its efficacy in the clinic remains limited primarily due to strict dose restrictions. Indeed, only 50% of patients with normal renal function are able to maintain concentrations of colistin in their serum at levels that are sufficient to eliminate bacteria[11,12]. Moreover, colistin is less effective when applied in vivo, since its efficacy often does not match results obtained by in vitro studies, with up to 70% of patients failing to respond to colistin treatment[13].

In addition to the constraints encountered in the clinical application of colistin, the emergence of multiple resistance mechanisms against this antibiotic further limits its efficacy during the treatment of recalcitrant infections. Colistin is a polycationic peptide that acts on Gram-negative bacteria by binding to the anionic lipid A component of the lipooligosaccharides (LOS) in the outer membrane, as well as the lipid A found in the inner membrane[14,15]. This interaction between colistin and lipid A has been shown to induce membrane permeabilization and cell leakage[16]. This process is accompanied by an increase in the production of reactive oxygen species (ROS) within the bacterial cell, which ultimately leads to cell death[17]. Resistance to colistin in *A. baumannii* is primarily associated with modifications to or complete loss of lipid A[18–23]. These changes are driven by a range of enzymes that are capable of modifying lipid A, as well as different regulators, which control the expression of these enzymes[24–29]. The majority of colistin-resistant strains harbour mutations in the two-component system coding genes *pmrA* or *pmrB,* which trigger increased expression of the gene encoding the phosphoethanolamine transferase, PmrC[30]. When overexpressed, PmrC adds phosphoethanolamine (PEtN) to lipid A at the 1' or 4' phosphate position, thus reducing the overall negative surface charge of the bacterial cell and decreasing the binding of colistin and reducing its killing efficacy[31]. Mutations in the *pmrB* locus can also cause the overexpression of NaxD, which modifies the lipid A with positively charged galactosamine (GalN) and, similarly to PEtN, decreases the negative surface charge leading to the repulsion of colistin[32,33]. Other members of the phosphoethanolamine transferase enzyme family, like the recently discovered mobile colistin resistance (MCR) enzymes[34,35] and endogenous EptA proteins[30,36], perform the same PetN modification to the lipid A when acquired or upregulated, respectively. Finally, colistin resistance can also occur through complete loss of LOS in *A. baumannii*, through the inactivation of any of the first three enzymes involved in the Raetz pathway, LpxA, LpxC or LpxD[22,37–41].

The lack of treatment options against carbapenem-resistant *A. baumannii* due to the multiplicity of colistin resistance mechanisms in this organism, along with the aforementioned limitations during clinical administration of polymyxin antibiotics, have generated an urgent need for the discovery of novel strategies that can increase the efficacy of colistin or sensitise colistin-resistant strains. To this aim, phytochemicals are a promising reservoir of compounds with desirable properties for the generation of novel antibacterial agents. They have been shown to possess a far greater structural diversity and complexity in comparison to synthetic compound libraries and have superior ADME/T (absorption, distribution, metabolism, excretion, and toxicity) properties[42]. This is likely because plants have endured millennia of selective pressure that has geared them towards developing effective small molecules to prevent colonisation by bacterial pathogens[43].

Here we perform high throughput screening of phytochemical libraries and identify kaempferol as a next-generation antimicrobial. When applied as a monotherapy, kaempferol inhibits biofilm formation without affecting bacterial growth, while in combination with colistin, it potentiates the activity of this polymyxin compound against multidrug-resistant (MDR) *A. baumannii* AB5075. We show that colistin potentiation is due to the disruption of intracellular iron homeostasis and exposes a previously unrecognised metabolic vulnerability of *A. baumannii*, whereby even sub-inhibitory concentrations of colistin are sufficient to cause detrimental metabolic re-wiring. Finally, we demonstrate that this vulnerability can be therapeutically exploited more broadly, since the combination of colistin and kaempferol overcomes colistin resistance in critical Gram-negative bacterial pathogens.

## Results

**Kaempferol is a potentiator of the antimicrobial activity of colistin.** To identify potentiators of colistin activity, a library of phytochemicals was screened in the presence of a sub-inhibitory concentration of colistin (1.22 µg/mL) against an MDR clinical isolate, *A. baumannii* AB5075. We concurrently ran a control screen in the absence of colistin to exclude phytochemicals with colistin-independent antimicrobial effects on the growth of *A. baumannii*. From the initial screening, we found that kaempferol, a natural flavanol flavonoid that can be isolated from many plant-derived foods like strawberries and capers[44], had the most potent and consistent antimicrobial effect when combined with sub-MIC amounts of colistin in vitro. Titration assays revealed that this effect is dose-dependent and that a concentration of 0.375 mM of kaempferol combined with 1.22 µg/mL of colistin leads to complete inhibition of visible bacterial growth (Fig. 1a). We further determined that this concentration of kaempferol had no impact on *A. baumannii* growth when administered as a monotherapy (Fig. 1b), suggesting that kaempferol potentiates the activity of colistin without affecting bacterial growth on its own.

**Kaempferol is a better potentiator than its derivatives.** To explore the relationship between the chemical structure of kaempferol and its promise as a colistin potentiator, we tested a panel of kaempferol derivatives (Fig. 2a). The native kaempferol structure was found to be the most efficacious at reducing the growth of *A. baumannii*, when combined with sub-MIC amounts of colistin (Fig. 2b). In particular, we found that there was an inverse relationship between the extent of chemical modifications applied to these compounds and their bactericidal activity in combination with colistin. For example, heavily modified derivatives such as kaempferol 3-glucoside 4',7-dirhamnoside were unable to potentiate colistin activity, while compounds with minor modifications (5-Deoxykaempferol, also known as Resokaempferol) acted as colistin potentiators.

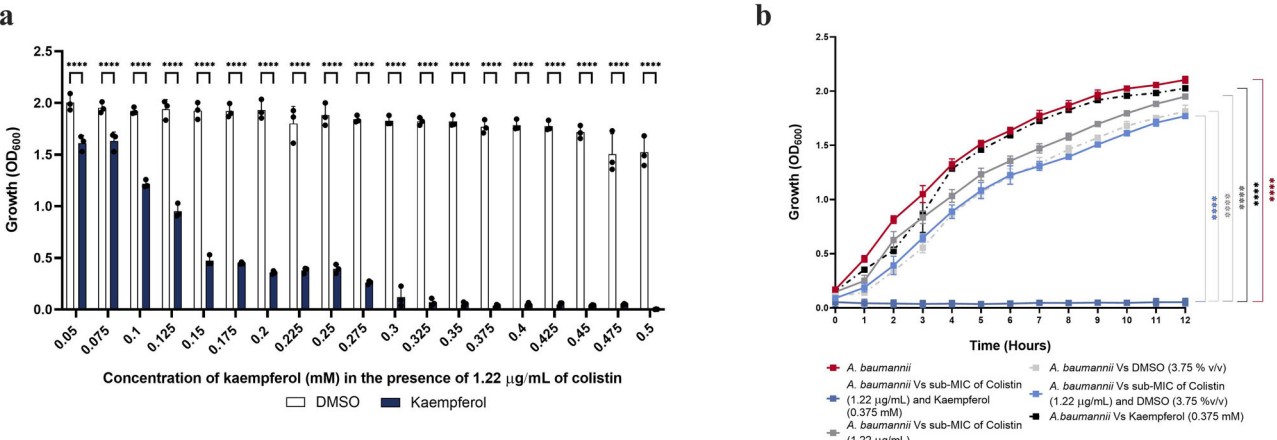

**Fig. 1 Kaempferol potentiates the activity of colistin. a** Minimum inhibitory concentration (MIC) of the kaempferol-colistin combination treatment. Complete inhibition of visible growth of *A. baumannii* was observed when 0.375 mM of kaempferol were combined with sub-MIC amounts of colistin (1.22 μg/mL). Inhibition was shown to be dose dependent. **b** Inhibition of *A. baumannii* growth by kaempferol in combination with sub-MIC amounts of colistin. Monotherapy with 1.22 μg/mL of colistin or 0.375 mM of kaempferol did not inhibit growth. For both panels, assay results were carried out in biological triplicate, each performed in technical triplicate. Statistical analysis consisted of two-way ANOVA for (**a**) and two-way repeated measures ANOVA for (**b**) between the treated samples and the respective DMSO growth controls. Average values ± S.D. are represented. Significance is indicated as *$p \leq 0.05$, **$p \leq 0.01$, ***$p \leq 0.001$, ****$p \leq 0.0001$.

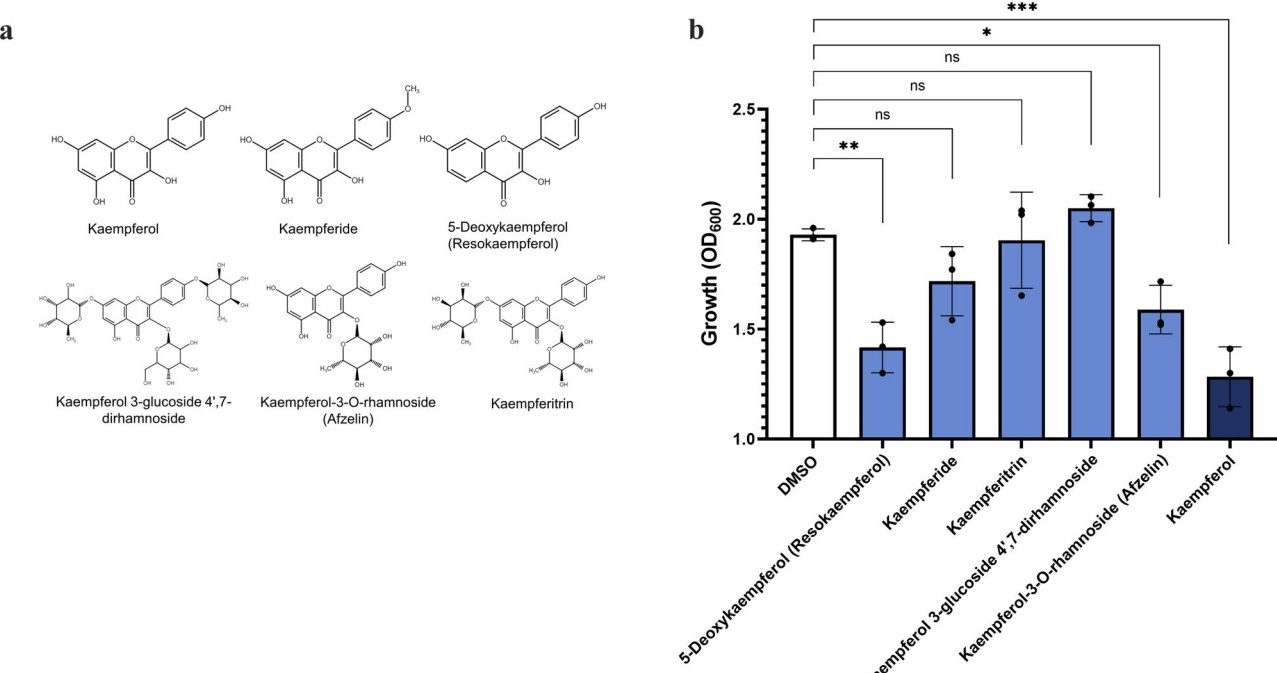

**Fig. 2 Kaempferol and structurally related compounds potentiate the antimicrobial activity of colistin. a** Chemical structures of kaempferol and kaempferol-like molecules. **b** Impact of kaempferol derivatives at a concentration of 0.05 mM on the growth of *A. baumannii*. The native structure of kaempferol has the highest potentiating activity against *A. baumannii* when combined with sub-MIC amounts of colistin (1.22 μg/mL). For (**b**), assays were carried out in biological triplicate, with three technical repeats. Statistical analysis consisted of one-way ANOVA between the treated sample and the DMSO carrier control. Average values ± S.D. are represented. Significance is indicated as ns = non-significant, *$p \leq 0.05$, **$p \leq 0.01$, ***$p \leq 0.001$, ****$p \leq 0.0001$.

**Kaempferol has antibiofilm activity when applied as a monotherapy.** Flavonoids have been previously reported to have activity against biofilm formation[45] and, more specifically, kaempferol has been shown to inhibit *Staphylococcus aureus* biofilms[46]. Given the well-established link between bacterial antibiotic tolerance and biofilm formation[47], we wanted to explore whether kaempferol, or its derivatives, have antibiofilm potential. We found that the native structure of kaempferol is

active against the formation of *A. baumannii* biofilms, and heavily modified structures such as kaempferol 3-glucoside 4',7-dirhamnoside had even higher antibiofilm activities (Fig. 3a).

Considering that heavily modified derivatives, such as kaempferol 3-glucoside 4',7-dirhamnoside, showed almost no colistin potentiation (Fig. 2b), these results suggest that the mechanism through which biofilm formation is inhibited is independent to that of mediating colistin potentiation. We further

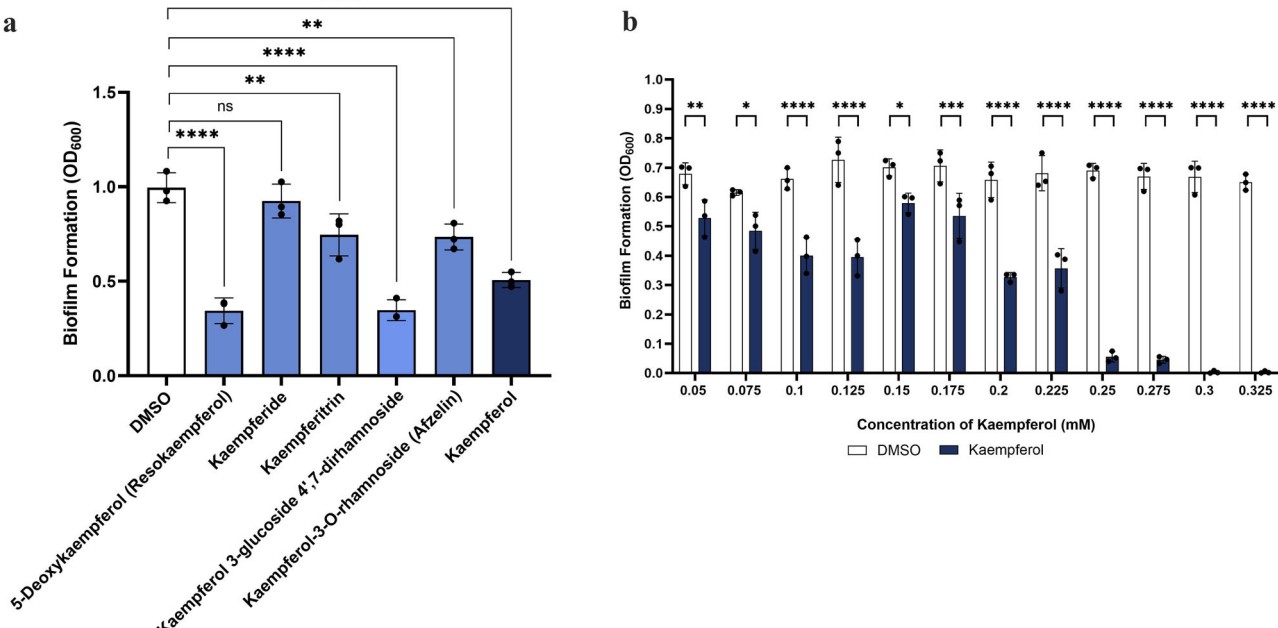

**Fig. 3 Kaempferol and structurally related compounds inhibit biofilm formation. a** Impact of kaempferol-like compounds at a concentration of 0.05 mM on *A. baumannii* biofilm formation. The native structure of kaempferol, along with some of its derivatives, have activity against *A. baumannii* biofilms when applied as a monotherapy. **b** Minimum biofilm inhibitory concentration (MBIC) of kaempferol on *A. baumannii*. Complete biofilm inhibition was observed at a concentration of 0.3 mM, and inhibition is shown to be dose dependent. For all panels, assays were carried out in biological triplicate, with three technical repeats. Analysis consists of one-way ANOVA for (**a**) and two-way ANOVA for (**b**), between the treated samples and the DMSO carrier control. Significance is indicated as ns = non-significant, *$p \leq 0.05$, **$p \leq 0.01$, ***$p \leq 0.001$, ****$p \leq 0.0001$.

explored the antibiofilm formation capacity of the native structure of kaempferol by performing a minimum biofilm inhibitory concentration (MBIC) assay. We found that kaempferol has a dose-dependent impact on biofilm formation; lower concentrations (<0.05 mM), as previously reported, have minimal effects on biofilm formation[48], while higher concentrations (0.3 mM) reduce biofilm by >90% compared to the carrier control, without impacting bacterial growth (Fig. 3b). These findings suggest that kaempferol and some of its derivatives can be classified as next-generation antimicrobials i.e., compounds that do not inhibit growth on their own, but can inhibit processes associated with virulence and infection progression.

**Kaempferol affects iron homeostasis in *A. baumannii* AB5075**. We started investigating the mechanism by which kaempferol promotes colistin potentiation in *A. baumannii* by performing a differential RNA sequencing (dRNA-seq) experiment. We treated cells with kaempferol at a concentration where it is most effective when combined with colistin (0.375 mM) and compared the gene expression profile to cells only exposed to the DMSO carrier control. We found that 117 genes were differentially expressed at significant levels, with 99 being upregulated and 18 downregulated (Fig. 4). We further analysed our dRNA-seq data by using the current *A. baumannii* AB5075-UW annotated genome[49] as a reference and performing Gene Set Enrichment Analysis with FUNAGE-Pro[50] to obtain more information about the functionality of the differentially regulated genes (Supplementary Data 1). Strikingly, numerous genes related to siderophore biosynthesis and transport in *A. baumannii* AB5075 were upregulated, including the acinetobactin (*bar-bas-bau* cluster, ABUW_1168-1188) and bauminoferrin (*bfn* gene cluster, ABUW_2178-2189) biosynthetic pathways. In addition, two bacterioferritin orthologues (*bfr*, ABUW_0306 and *bfrA*, ABUW_3125) that likely function as iron storage proteins, were

found to be downregulated. Together these data suggest that kaempferol is affecting the iron homeostasis in AB5075, and more specifically, indicates that cells are suffering from low iron levels in the presence of this compound. Our dRNA-seq data suggests that kaempferol acts as an iron chelator. We confirmed this by measuring the shift in the spectroscopic absorption peak of kaempferol in the presence and absence of iron (Supporting Information Fig. S1), which agrees with previous work[51].

**Kaempferol-induced iron dysregulation drives colistin potentiation**. According to our transcriptomic data (Fig. 4 and Supplementary Data 1), the presence of kaempferol affects iron homeostasis in treated *A. baumannii* cells. To investigate whether this kaempferol-induced dysregulation of cellular iron content underpins colistin potentiation, we supplemented cells grown in the presence of colistin and kaempferol with $Fe^{3+}$ or $Fe^{2+}$. Strikingly the increase in bioavailability of $Fe^{3+}$ reversed the previously observed growth defect (Fig. 5a). In agreement with our spectroscopic assays, the addition of $Fe^{2+}$ did not rescue growth (Fig. 5b). These data suggest that $Fe^{3+}$ homeostasis is critical during exposure of bacteria to colistin, even at subinhibitory concentrations. To further demonstrate the importance of iron homeostasis during colistin treatment, we exposed cells to a non-specific iron chelator, EDTA, in the presence of a subinhibitory concentration of colistin[52]. Similar to kaempferol, EDTA potentiated the activity of colistin (Fig. 5c), thus highlighting the key role of iron homeostasis for bacteria that are trying to overcome colistin exposure. In addition, we used specific iron chelators, ExJade and 8-hydroxyquinoline[53,54], and found that they could also potentiate the activity of colistin, supporting our hypothesis that it is iron chelation by kaempferol that mediates killing of *A. baumannii* in sub-inhibitory amounts of colistin (Supporting Information Fig. S2). Finally, we measured the levels of intracellular $Fe^{3+}$ and $Fe^{2+}$ in the presence of kaempferol and

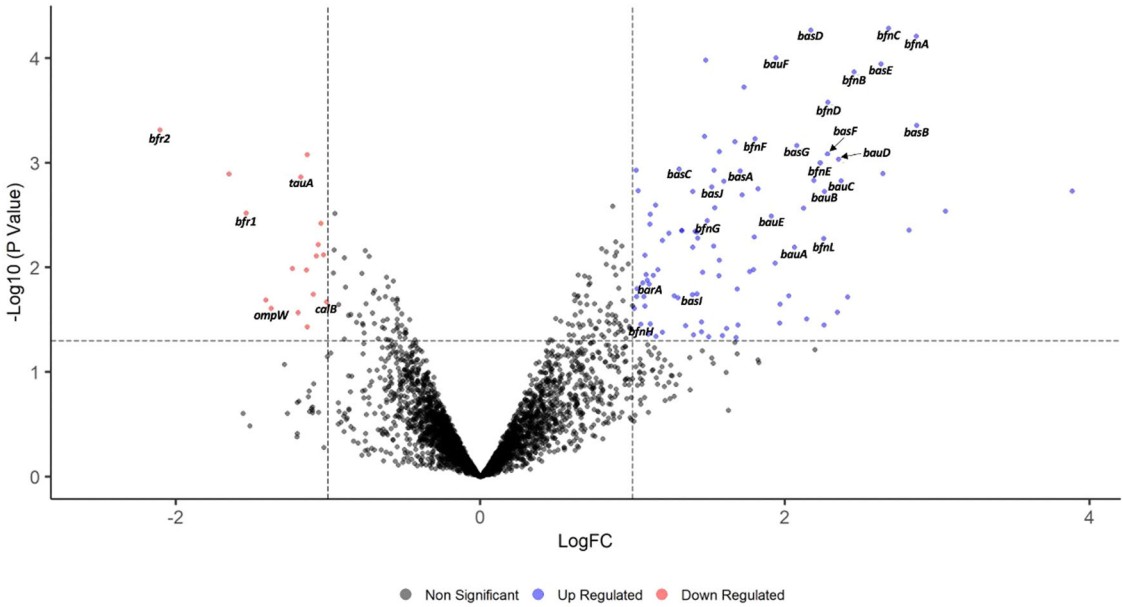

**Fig. 4 Kaempferol affects iron homeostasis in *A. baumannii* AB5075.** Volcano plot representing the results of the dRNA-seq analysis performed on cells exposed to kaempferol or the DMSO carrier control. Genes with no change in their expression levels are shown in black, whereas significantly downregulated and upregulated genes are labelled in red and blue, respectively. A total of 117 genes were found to have significantly altered expression levels (99 upregulated and 18 downregulated). Representative cases are indicated, for example upregulated genes related to siderophore biosynthesis and transport pathways, such as acinetobactin (*bar-bas-bau* cluster) and bauminoferrin (*bfn* gene cluster), as well as downregulated genes related to iron storage (bacterioferritin orthologues *bfr1* and *bfr2*) or encoding iron-dependent proteins (*calB*, *tauA*, and *ompW*).

observed a significant decrease compared to the carrier control (Fig. 5d), further confirming the impact of kaempferol on iron availability within the cell.

**Kaempferol does not increase membrane permeability.** In a contemporaneous report[48], the potentiation of colistin by kaempferol was also reported, and it was proposed that the potentiation of colistin by kaempferol against *A. baumannii* relies on increased membrane permeability. To explore this hypothesis, we investigated the effect of the kaempferol-colistin combination on the integrity of the outer and inner membranes of *A. baumannii*. We found that there was no effect on the integrity of the outer membrane when cells were treated with the combination, kaempferol or colistin alone at the experimental concentrations that we have used throughout our study. By contrast, significant 1-N-phenylnapthylamine (NPN) uptake was observed when treated with a high concentration of colistin (16 μg/mL) (Fig. 6a). Notably, the presence of kaempferol, either alone or in combination with colistin, resulted in significantly decreased permeability. Similarly, we did not record any inner membrane permeabilization when treated with either the combination, kaempferol or colistin alone at the same concentrations, whereas significant propidium iodide (PI) uptake was observed when treated with a high concentration of colistin (16 μg/mL) (Fig. 6b). Together, these results suggest that neither of the two components of our combination treatment (colistin or kaempferol) applied at the concentrations used in this study compromise the integrity of the cell envelope and that there is an alternative primary mechanism of action for colistin potentiation by kaempferol.

**Kaempferol in combination with colistin causes a lethal accumulation of ROS.** Given that the cell envelope integrity of *A. baumannii* is not affected by combination treatment with kaempferol and colistin, we explored other potential mechanisms

through which potentiation might be occurring. One of the known mechanisms of action of colistin against *A. baumannii* is the hydroxyl radical death pathway, whereby superoxide ($O_2^-$) is generated after colistin transits through the outer membrane and crosses the inner membrane[17]. Superoxide is then detoxified into hydrogen peroxide ($H_2O_2$) by superoxide dismutase (SOD) enzymes in the cytoplasm, and the generated $H_2O_2$ subsequently oxidises $Fe^{2+}$ into $Fe^{3+}$[55,56]. This conversion is designated Fenton's reaction and, together with the Haber-Weiss reaction, is a main source of reactive oxygen species (ROS) in the cell[57,58]. Considering that kaempferol impacts iron homeostasis (Figs. 4, 5), we posited that its combination with colistin might affect the downstream production of ROS. We started investigating this hypothesis by measuring total ROS production in cells exposed to kaempferol, colistin or both. We found that sub-inhibitory concentrations of colistin induced ROS production, while kaempferol alone did not (Fig. 7a). In agreement with our hypothesis, the combination treatment induced levels of ROS that significantly exceeded those induced by colistin alone. Further supporting this, the amount of ROS produced by colistin alone was not sufficient to affect growth (Fig. 7a, Supporting Information Fig. S3). Together, this suggests that the ROS detoxification systems that would normally rescue the cells from exposure to colistin at sub-MIC amounts, may be impaired in the presence of kaempferol, in turn leading to the accumulation of ROS at levels that are lethal for the cell.

In *A. baumannii*, the only annotated SOD coding genes are *sodB* (ABUW_1216) and *sodC* (ABUW_0339). Intriguingly, these enzymes require metal cofactors for activity, with SodB binding iron, and SodC depending on copper or zinc. To investigate whether either of these enzymes are essential for the detoxification of ROS produced as a result of colistin exposure, we treated transposon mutants in the respective coding genes obtained from the Manoil *A. baumannii* transposon mutant library[49] with colistin at sub-MIC amounts. Both mutants grew similarly to the wild-type strain when exposed to the carrier control or to

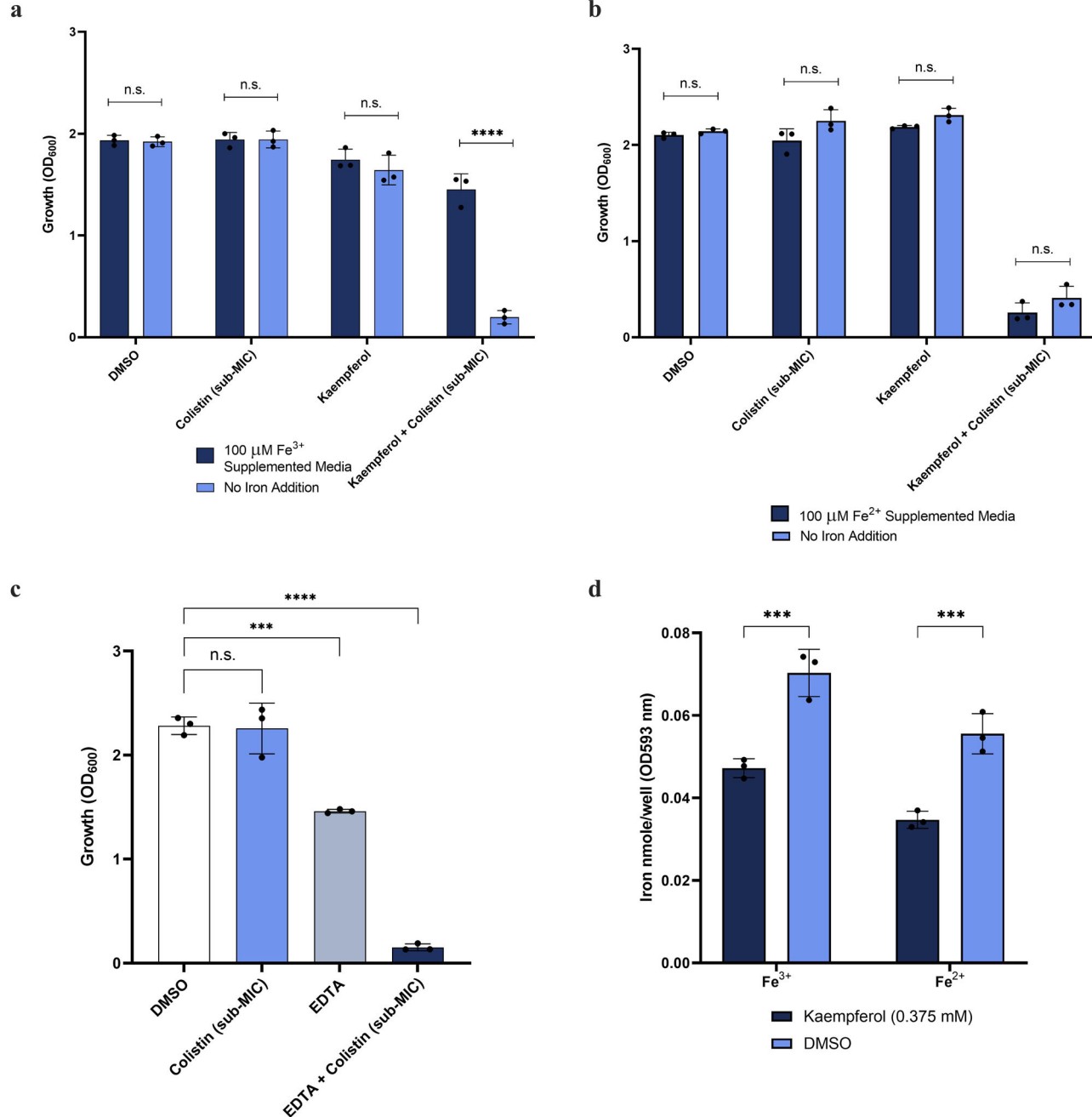

**Fig. 5 Dysregulation of iron homeostasis underpins colistin potentiation by kaempferol.** Growth inhibition of *A. baumannii* in the presence of 0.375 mM kaempferol concentration in combination with sub-MIC amounts of colistin (1.22 μg/mL) is rescued by $Fe^{3+}$ (**a**), but not $Fe^{2+}$ (**b**) supplementation. **c** EDTA at 100 μM potentiates the activity of colistin; colistin was used at sub-MIC amounts (1.22 μg/mL). **d** The concentration of available intracellular $Fe^{3+}$ and $Fe^{2+}$ was decreased in the presence of the kaempferol, compared to the DMSO control. For all panels, assays were carried out in biological triplicate, with three technical repeats. Analysis consists of independent *t* test for (**a**, **b** and **d**) and one-way ANOVA for (**c**), between the treated samples and the DMSO carrier control. Average values ± S. D. are represented. Significance is indicated as ns = non-significant, *$p \leq 0.05$, **$p \leq 0.01$, ***$p \leq 0.001$, ****$p \leq 0.0001$.

kaempferol alone (Fig. 7b), highlighting that dysregulation of iron homeostasis under normal growth conditions i.e., conditions where excess ROS is not generated, is not detrimental to the cell. Nonetheless, when colistin was used at sub-MIC levels, the growth of the *sodB* mutant was completely impaired, while the *sodC* mutant growth was only marginally affected (Fig. 7b).

With SodB requiring iron as a cofactor for it's activity, it is possible that the potential decrease of available iron in the presence of kaempferol, in addition to affecting Fenton's reaction,

also affects SodB function. This would further decrease the ability of the cell to detoxify the excess ROS produced in the presence of colistin, even at sub-MIC amounts (Fig. 7a)[59]. In order to investigate whether the iron-dependency of SodB contributes to the potentiation of colistin by kaempferol, we constructed two wild-type derivative strains overexpressing either *sodB* or *sodC* under an IPTG-inducible promoter and challenged them with our kaempferol and colistin combination treatment. We hypothesised here that if iron was essential for SodB activity, then

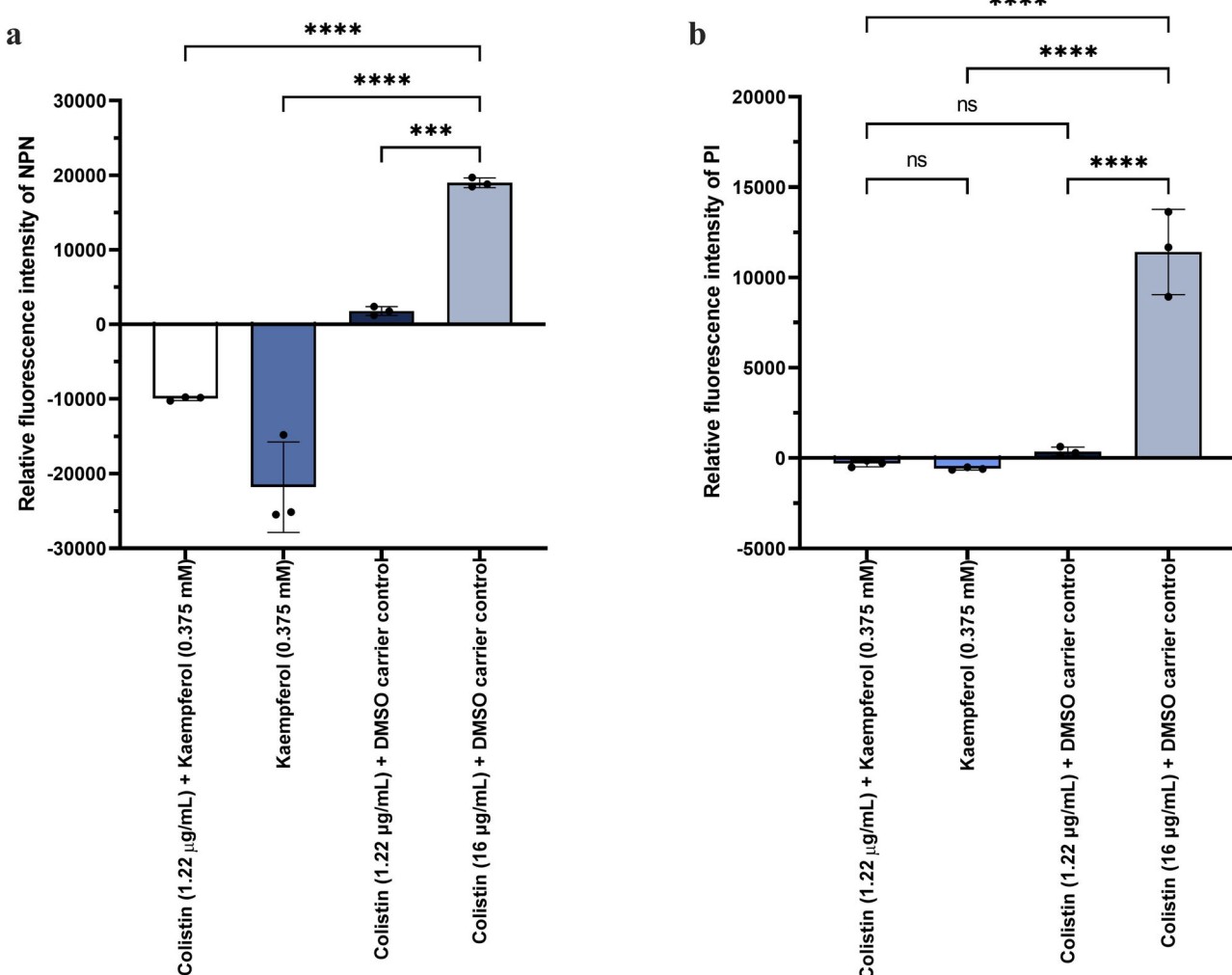

**Fig. 6 Kaempferol-colistin combination treatment or its individual components do not increase membrane permeability in *A. baumannii*. a** Combination of sub-MIC amounts of colistin (1.22 μg/ml) and kaempferol (0.375 mM), or kaempferol (0.375 mM) or colistin (1.22 μg/ml) alone, do not increase NPN uptake in *A. baumannii* cells, and thus do not increase the permeability of the outer membrane. High amounts of colistin (16 μg/mL) permeabilize the outer membrane. **b** Combination of sub-MIC amounts of colistin (1.22 μg/ml) and kaempferol (0.375 mM), or kaempferol (0.375 mM) or colistin (1.22 μg/ml) alone, do not increase PI uptake in *A. baumannii* cells, and thus do not increase the permeability of the inner membrane. High amounts of colistin (16 μg/mL) permeabilize the inner membrane. All assays were carried out in biological triplicate, with three technical repeats. For both panels, DMSO (the kaempferol carrier) was added to all experiments where cells were treated with colistin only. Analysis consisted of one-way ANOVA between the treated samples and the colistin (16 μg/mL) control. Average values ± S. D. are represented. Significance is indicated as ns non-significant, *$p \le 0.05$, **$p \le 0.01$, ***$p \le 0.001$, ****$p \le 0.0001$.

overexpressing *sodB* in the presence of the combination treatment would not rescue the cells, however, overexpression of *sodC*, which does not bind iron, should be beneficial. Indeed, our assays confirmed that this is the case, with only *sodC* overexpression partially rescuing the lethal phenotype of the kaempferol and colistin combination treatment (Fig. 7c). This suggests that colistin potentiation in the presence of kaempferol is affected by the abundance and enzymatic activity of the different SOD enzymes. To further explore this, we generated *gfpmut3* transcriptional fusions to the *sodB* and *sodC* promoter regions and the expression of these genes in the presence and absence of the combination treatment was determined. As shown in Fig. 7d, the levels of expression of both genes remained invariant across the combination treated samples and DMSO controls, highlighting that the effects of kaempferol are not occurring at the transcriptional level, but at the enzymatic activity level of SOD proteins. Our results in Fig. 7d also confirm our dRNA-Seq data, where greater levels of *sodB* expression were measured compared

to those of *sodC*. These observations, together with the fact that overexpression of *sodC* only partially rescued the lethal phenotype of the combination treatment (Fig. 7c), further highlight the importance of SodB in ROS detoxification. To assess whether other metal cofactors or cations can overcome the influence of kaempferol, we supplemented the media with $Cu^{2+}$, $Mg^{2+}$, $Zn^{2+}$ or $Ca^{2+}$ and found that they could not rescue growth inhibition from the combination treatment (Fig. 7e). Altogether, these results indicate that the iron sequestration by kaempferol disables the ROS protection mechanisms of *A. baumannii*, which cannot be overcome by iron-independent mechanisms, leading to greater colistin susceptibility.

**Kaempferol sensitizes colistin-resistant clinical strains.** Colistin potentiation is critical for safeguarding this last resort antibiotic as it is often our only treatment option against highly resistant Gram-negative pathogens. We have shown that in *A. baumannii*

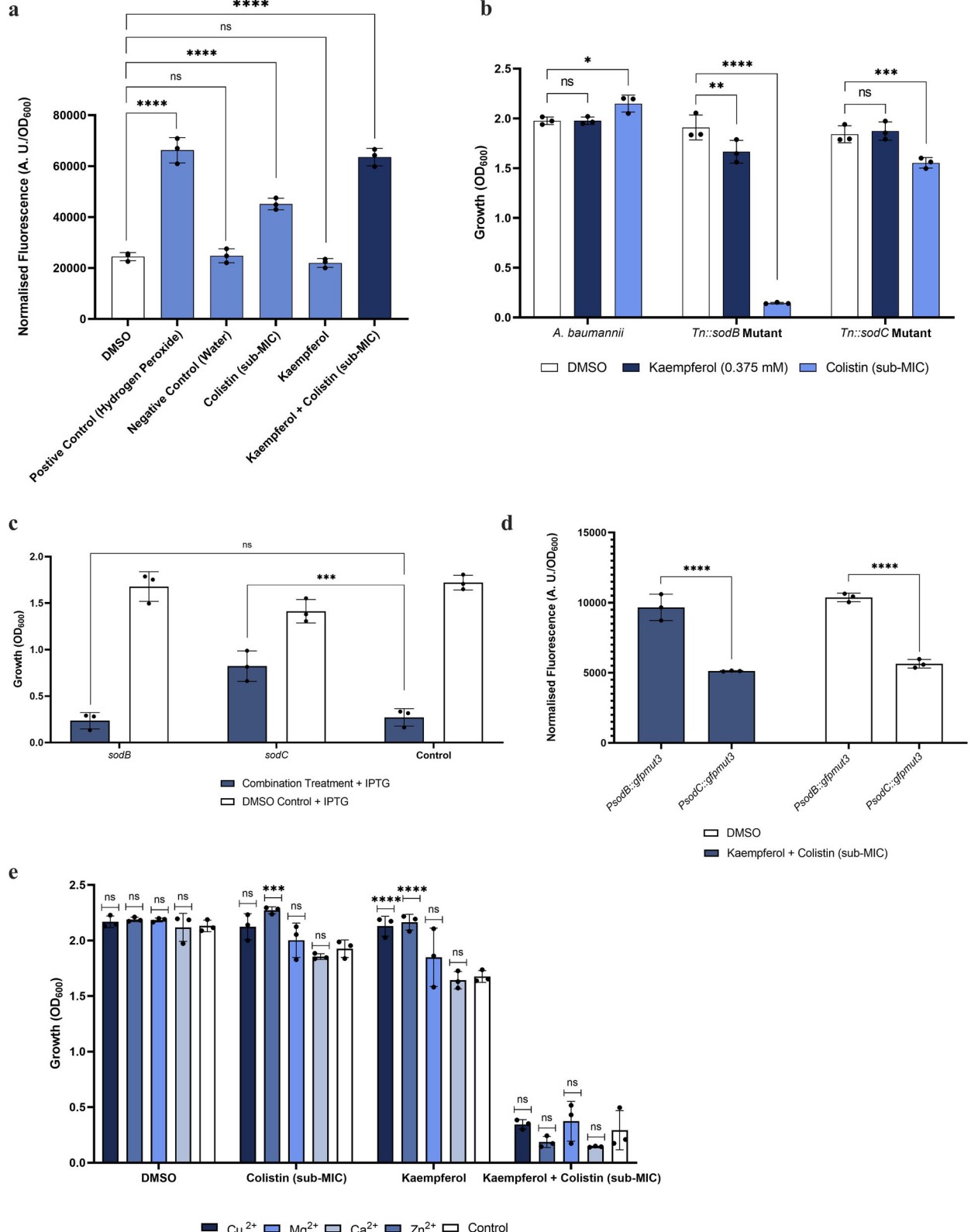

kaempferol chelates iron (Figs. 4, 5), increases cytoplasmic ROS production and prevents ROS detoxification during colistin treatment (Fig. 7). To assess whether these effects would be sufficient to increase the efficacy of colistin against strains that are resistant to this antibiotic, we first tested our potentiation approach on laboratory model strains. Since iron homeostasis and ROS detoxification are important for most bacteria[60,61] we

started our investigation using *Escherichia coli* K-12; we selected the W3110 strain, along with its colistin-resistant counterparts, strain WD101, a *pmrA* mutant with 4-amino-4-deoxy-L-arabi-nose (L-Ara4N)-modified lipid A[62], and strain W3110, carrying the *mcr-1* gene on a medium copy-number plasmid that bears PEtN-modifications on its lipid A[63]. The use of these strains allowed us to assess our approach on a well-controlled system

**Fig. 7 Kaempferol inhibits ROS detoxification during colistin treatment. a** Sub-MIC amounts of colistin (1.22 μg/ml) induce ROS production, while kaempferol alone (at 0.375 mM) does not. The kaempferol and colistin combination treatment (colistin at 1.22 μg/ml and kaempferol at 0.375 mM) induces levels of ROS that significantly exceed those induced by colistin alone. The positive control consisted of hydrogen peroxide at a concentration lethal to *A. baumannii* (10% v/v) (Supporting Information Fig. S3). **b** Growth of *sodB* and *sodC* transposon mutants in the presence of DMSO, kaempferol, colistin and the combination of kaempferol and colistin (colistin was used at 1.22 μg/ml and kaempferol at 0.375 mM). The *sodB* mutant is more susceptible to sub-MIC amounts of colistin compared to wild-type *A. baumannii* or the *sodC* mutant. **c** Growth of AB5075 derivative strains overexpressing either *sodB* or *sodC* from a miniTn7-based IPTG-inducible system compared to an empty-vector control. The overexpression of *sodB* could not restore the growth after treatment with kaempferol and colistin (colistin was used at 1.22 μg/ml and kaempferol at 0.375 mM). However, the overexpression of *sodC* could significantly alleviate growth inhibition. Statistical comparisons were performed between the overexpression strains and the empty-vector control. **d** Fluorescence measurements from a transcriptional *gfp* fusion to the *sodB* and *sodC* promoter regions (P*sodB* and P*sodC*, respectively) after growth for 2 h in the presence of DMSO or the combination of kaempferol and colistin (colistin was used at 1.22 μg/ml and kaempferol at 0.375 mM). For both conditions (carrier control or combination treatment) *sodB* is expressed at higher levels than *sodC*. **e** Growth inhibition of *A. baumannii* in the presence of 0.375 mM kaempferol concentration in combination with sub-MIC amounts of colistin (1.22 μg/mL) is not rescued by $Cu^{2+}$, $Mg^{2+}$, $Zn^{2+}$ or $Ca^{2+}$ supplementation. All assays were carried out in biological triplicate, with three technical repeats. Analysis consisted of one-way ANOVA for (**a**) and two-way ANOVA for (**b**), comparing the treated samples with the DMSO carrier control. Two-way ANOVA analysis was carried out for (**c**) and (**e**) comparing the treated the supplemented samples and the $H_2O$ control. Two-way ANOVA analysis was carried out for (**d**) between P*sodB::gfpmut3* and P*sodC::gfpmut3*. Average values ± S.D. are represented. Significance is indicated as ns = non-significant, *$p \leq 0.05$, **$p \leq 0.01$, ***$p \leq 0.001$, ****$p \leq 0.0001$.

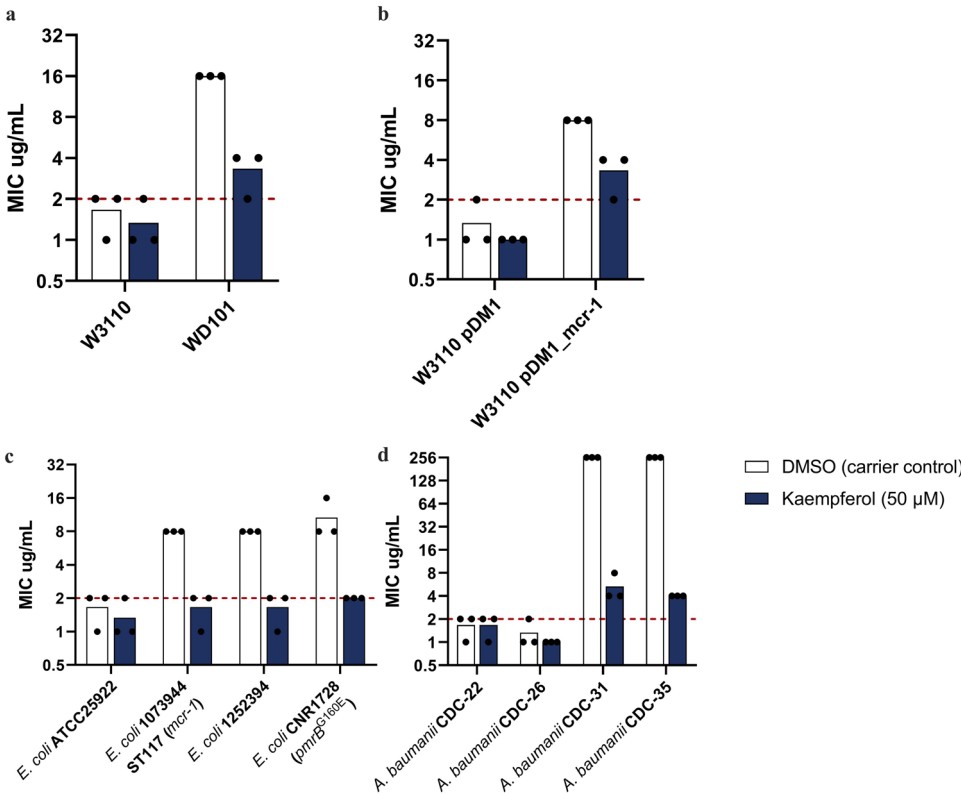

**Fig. 8 Kaempferol addition results in colistin potentiation when applied to clinical strains of *E. coli* and *A. baumannii*. a**, **b** Combination of colistin with kaempferol, results in decrease of colistin MIC values for colistin-resistant *E. coli* K-12 strains, independent of the mechanism of colistin resistance; sensitive strains remain unaffected. WD101 is a *pmrA* mutant with 4-amino-4-deoxy-L-arabinose (L-Ara4N)-modified lipid A derived from strain W3110, and W3110 pDM1_*mcr-1* expresses the mobile colistin resistance protein MCR-1 and, therefore bears PEtN-modifications on its lipid A. **c** Combination of colistin with kaempferol sensitises colistin-resistant *E. coli* clinical isolates. **d** Combination of colistin with kaempferol results in a 64-fold drop of the colistin MIC values of colistin-resistant *A. baumannii* clinical isolates. All assays were carried out in biological triplicate, with four technical repeats; red dotted lines indicate the EUCAST clinical breakpoint for colistin.

with defined lipid A modifications, devoid of confounding effects that might arise had we only tested non-isogenic clinical isolates. We compared the colistin MIC values for these strains recorded under EUCAST specifications in the presence of 50 μM kaempferol to ones obtained only in the presence of the DMSO carrier control. We found that for colistin-susceptible strain backgrounds (W3110 and W3110 carrying an empty-vector control (pDM1)), the addition of kaempferol did not have any major effects on their colistin MIC values (Fig. 8a, b). However, for strains with

chromosomal- (WD101) and plasmid-encoded (W3110+*mcr-1*) resistance, the use of kaempferol resulted in colistin MIC drops of 8 and 4 μg/mL, respectively (Fig. 8a, b). This demonstrates that colistin potentiation due to kaempferol exposure might be a useful avenue to reverse colistin resistance, irrespective of the mechanism of lipid A modification.

Encouraged by these results, we tested our approach on *E. coli* clinical isolates; we selected a colistin-susceptible *E. coli* (ATCC 25922, standardly used as a EUCAST MIC assay control),

a colistin-resistant *mcr-1*-carrying strain (1073944), and two colistin-resistant isolates harbouring chromosomal mutations (1252394 and CNR1728). For all three colistin-resistant isolates, the addition of kaempferol resulted in complete sensitization to colistin, while the sensitive strain remained unaffected (Fig. 8c). We then proceeded to investigate whether similar results could be obtained in *A. baumannii* clinical strains (Fig. 8d and Supplementary Data 2). We exposed highly resistant *A. baumannii* isolates (colistin MIC values of 256 µg/mL) to the combination treatment and observed a 64-fold drop in MIC values (colistin MIC values of 4 µg/mL; we note that the EUCAST breakpoint for colistin is 2 µg/mL). Although sensitization was not achieved for *A. baumannii* isolates, these drastic drops in colistin MIC values demonstrate the potential of kaempferol as a colistin potentiator, particularly in light of colistin's high toxicity and low bioavailability during treatment[11].

**Kaempferol promotes clearance of *A. baumannii* in vivo.** To further assess kaempferol's potential as a colistin potentiator, we tested our combination approach in vivo by using the *Galleria mellonella* wax moth model of infection[64,65]. Since the efficacy of antibiotics in vivo is often different from the MIC values reported in vitro[66], we first performed a titration for colistin in our infection model. We found that the lowest concentration of colistin allowing larval survival after lethal infection with *A. baumannii* is 0.1 µg/*G. mellonella* (Supporting Information Fig. S4). Therefore, we chose to use 0.08 µg of colistin/*G. mellonella*, a concentration that alone does not allow survival of the larvae (Supporting Information Fig. S4) in our subsequent experiments with kaempferol. We then assessed whether any of the compounds we intended to use had underlying toxicity in this model. Cytotoxicity assays (Supporting Information Fig. S5) showed that the DMSO carrier control, kaempferol and colistin at the amounts used were not toxic to the larvae, which have a probability of survival of >90% if exposed to any of these molecules. We finally tested the efficacy of our combination therapy approach in this infection model. We infected *G. mellonella* larvae with a lethal dose of *A. baumannii* and treated this infection with the carrier control (untreated), colistin at 0.08 µg/*G. mellonella* (monotherapy), kaempferol at 0.5 mM/*G. mellonella* (monotherapy), or with colistin at 0.08 µg/*G. mellonella* and kaempferol at 0.5 mM/*G. mellonella* (combination therapy). We found that if the infection remained untreated (carrier control) or was treated only with colistin, all of the larvae died (Fig. 9). Kaempferol monotherapy also resulted in less than 10% survival of the larvae. By contrast, combination therapy led to survival of 60% of the larvae, demonstrating that targeting iron bioavailability in an infection setting potentiates colistin activity.

**Discussion**
Kaempferol has been used in clinical trials for the treatment of diseases like acute and chronic inflammation, cancer, obesity, diabetes, and liver injury[44]. Studies have also shown that it has antimicrobial and antibiofilm activity against the Gram-positive bacteria *S. aureus* and *Enterococcus faecalis*[46,67,68]. Here, we demonstrate that the native form of kaempferol combined with sub-MIC amounts of colistin, is the most efficacious at reducing the growth of *A. baumannii* compared to its modified larger derivates (Fig. 2B, Supporting Information Figs. S6 and S7). We also demonstrate that the native form of kaempferol has anti-biofilm activity against the Gram-negative opportunistic pathogen *A. baumannii* and can potentiate the activity of colistin, bypassing resistance in *A. baumannii* and *E. coli* clinical strains. We found that the latter effect is, overall, due to the ability of kaempferol to disrupt intracellular iron homeostasis. Interestingly, a report

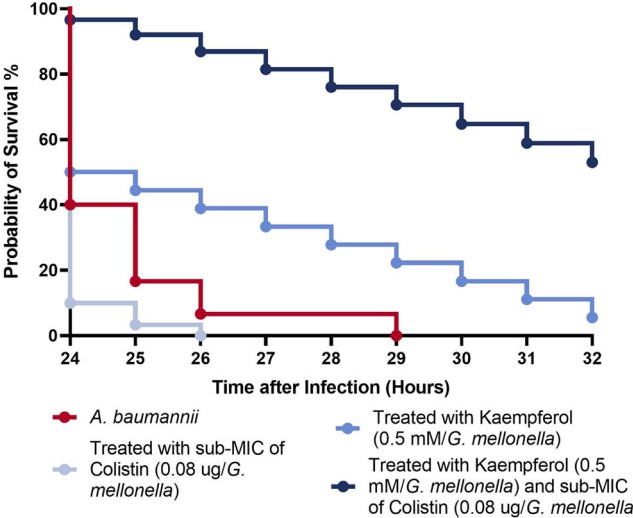

**Fig. 9 Kaempferol potentiates colistin activity in the *Galleria mellonella* infection model.** 60% of larvae survived *A. baumannii* infection when *G. mellonella* was treated with colistin at 0.08 µg/*G. mellonella* and kaempferol at 0.5 mM/*G. mellonella*. Lack of treatment (carrier control) and colistin monotherapy result in larval death, while kaempferol monotherapy allows less than 10% survival of the larvae. Assays were carried out in biological triplicate per treatment group, with 30 *G. mellonella* used per group ($N = 30$). Analysis consisted of Log rank (Mantel-Cox) test, where *A. baumannii* Vs *A. baumannii* + colistin was not significant, *A. baumannii* Vs *A. baumannii* + kaempferol was not significant and *A. baumannii* Vs *A. baumannii* + colistin + kaempferol had a Log Rank $p < 0.001$.

conducted concurrently with this study suggested that the combination treatment of kaempferol and colistin resulted in increased outer membrane permeability, as measured through alkaline phosphatase release after 6 hours[48]. However, our data did not support this hypothesis; by using membrane-specific dyes, we observed no increased inner membrane or outer membrane permeability caused by the concentrations of kaempferol or colistin used in this study (Fig. 6a, b).

While iron is an invaluable micronutrient that is critical for a wide array of metabolic functions in bacteria[69], free iron is toxic, and for this reason, its concentration is tightly regulated[70]. Our dRNA-Seq analysis (Fig. 4) showed that kaempferol exposure resulted in an upregulation of genes responsible for the biosynthesis and transport of the siderophores acinetobactin (*barbas-bau* cluster) and bauminoferrin (*bfn* gene cluster). At the same time, we observed a downregulation of two bacterioferritin orthologues (*bfr*) that promote iron storage, as well as genes encoding proteins that need iron as a cofactor (*calB* and *tauA*). Moreover, we recorded a downregulation of the *ompW* gene, a key component of the iron regulon that controls iron homeostasis[71]. Together, our transcriptomics data suggests that kaempferol disturbs intracellular iron homeostasis in *A. baumannii*, and in particular, drives the cell to acquire additional extracellular iron via increased siderophore production and decreased iron storage and usage. While disturbing iron homeostasis is stressful for the cell, it is clear from our iron rescue data that in rich media it can be overcome through these diverse responses. Nonetheless, during stress conditions, for example under colistin treatment, kaempferol-induced iron depletion becomes lethal, thereby potentiating the activity of colistin (Fig. 1a).

In agreement with our dRNA-Seq and absorbance spectra (Fig. 4 and Supporting Information Fig. S1), as well as previous reports[51,72] kaempferol binds to $Fe^{3+}$. The role of kaempferol in

iron chelation, and ultimately depletion, is further supported by the decrease of available intracellular iron in the presence of kaempferol (Fig. 5d), and the fact that supplementation of the growth media with $Fe^{3+}$ rescues the cells from the lethal effects of the kaempferol and colistin combination treatment (Fig. 5a). The importance of iron homeostasis for overcoming colistin treatment, which explains the lethal effects of the kaempferol and colistin combination treatment, is demonstrated by the fact that non-specific and specific iron chelators (EDTA, ExJade and 8-hydroxyquinoline) also potentiate the activity of colistin (Fig. 5c and Supporting Information Fig. S2).

We delved deeper into the mechanism of colistin potentiation by investigating whether kaempferol-induced iron imbalance in combination with colistin treatment led to the enhancement of the hydroxyl radical death pathway, a known mechanism of colistin against *A. baumannii*. In this pathway, $O_2^-$ generated due to colistin traversing through the outer and inner membrane is converted into $H_2O_2$ by SOD enzymes. $H_2O_2$ can then participate in the Fenton reaction to oxidise $Fe^{2+}$ into $Fe^{3+}$, concomitantly producing $\cdot OH$ (R2, below). Considering our data and the previous literature[51] that implicate kaempferol in chelating ferric iron thus making it unavailable, its presence could imbalance Fenton's reaction 1 and 2 (R1 and R2 below). Since kaempferol on its own does not have a growth inhibitory effect nor increases ROS production or damage the membrane (Fig. 1b, Fig. 6 and Fig. 7a, respectively), the cell is able to compensate for this imbalance when kaempferol is applied as a monotherapy. However, this imbalance is exacerbated by the toxic superoxide ($O_2^-$) generated in the presence of colistin[17,73], as shown by the increased ROS production during combination treatment (Fig. 7a). Since iron is a cofactor for SodB, a protein that is key for the detoxification of ROS during colistin treatment (Fig. 7b), chelation of iron by kaempferol further reduces the ability of the cell to detoxify the superoxide produced by colistin in the hydroxyl radical pathway. Overall, the synergy between colistin-induced ROS production and kaempferol-mediated disruption of the Fenton's reaction results in a dramatic increase of ROS species, whilst also rendering the cell unable to detoxify them, thereby having lethal consequences (Fig. 10).

$$\text{(Fenton's Reaction)} \quad Fe^{3+} + O_2^{\cdot} < - > Fe^{2+} + O_2 \quad \text{(R1)}$$

$$\text{(Fenton's Reaction)} \quad Fe^{2+} + H_2O_2 < - > Fe^{3+} + \cdot OH + H^+ \quad \text{(R2)}$$

$$\text{(Haber-Weiss Reaction)} \quad O_2^{\cdot} + H_2O_2 < - > \cdot OH + H^+ + O_2 \quad \text{(R3)}$$

In addition to elucidating the mechanism of the synergy between kaempferol and colistin, we demonstrate that the colistin potentiation activity of iron-chelating compounds can be exploited to reverse intrinsic and acquired resistance mechanisms (Fig. 8a, b). This approach can be used for highly colistin-resistant *A. baumannii* clinical strains (Fig. 8d), demonstrated by a staggering 64-fold drop in colistin MIC was observed. Moreover, in agreement with Zhou et al.[48], the combination of kaempferol with colistin, at amounts of colistin that would normally not rescue the *G. mellonella* larvae from a lethal dose of *A. baumannii*, results in 60% survival of the animals (Fig. 9). This shows that kaempferol can also be used to reduce the therapeutic dose of colistin. With consideration of the toxicity of colistin and the co-morbidities of patients that typically undergo this treatment, the kaempferol and colistin combination may significantly improve patient outcomes. In addition, colistin in combination with kaempferol is effective against colistin-resistant *E. coli* clinical strains. Isolates expressing MCR-1, as well as strains with chromosomal mutations causing colistin resistance, were sensitized against colistin in the presence

of kaempferol, highlighting the promise of this next-generation antimicrobial.

Overall, our study unveils a previously underappreciated metabolic vulnerability of bacterial pathogens i.e., their reliance on iron homeostasis for overcoming the action of colistin. Kaempferol disrupts the delicate balance of ROS production and detoxification that takes place during colistin treatment with detrimental effects to the cell. Beyond the promise that this compound holds as a colistin potentiator, our work opens new avenues towards the potentiation of colistin via molecules that broadly target processes and pathways whose role is to keep the ROS levels in the cell balanced.

## Materials and methods

**Bacterial strains**. *A. baumannii* (virulent colony variant)[74] and *E. coli* strains where routinely grown in LB media (Miller), either solid or liquid, static or shaking (180 rpm), respectively, at 37 °C. *A. baumannii* strain AB5075 used throughout this study was sourced from the Manoil Lab (University of Washington, Seattle, USA), with a colistin MIC of 2 μg/ml (Supporting Information Fig. S6). *A. baumannii* and *E. coli* strains, plasmids and oligonucleotides used in this study are listed in Supplementary Data 2, Supplementary Table S1 and Supplementary Table S2, respectively.

**Plasmid and strain construction**. In order to overexpress *sodB* and *sodC*, AB5075 miniTn7T-Tc derivative strains carrying those genes under the *lacIq-Ptac* expression system were generated. The plasmid pUC18T-miniTn7T-Tc-lacIq-Ptac was used as backbone for cloning into miniTn7T-Tc[75]. A DNA fragment containing the *sodB* coding region was PCR amplified from AB5075 genomic DNA using oligonucleotides sodB fw RBS PtsI and sodB rv KpnI. The *sodB* fragment was digested with PstI and KpnI and cloned into pUC18T-miniTn7T-Tc-lacIq-Ptac cut with the same enzymes. Similarly, a fragment containing *sodC* was amplified using primers sodC fw RBS PstI and sodC rv HindIII, digested with PstI and HindIII and cloned into pUC18T-miniTn7T-Tc-lacIq-Ptac cut with the same enzymes. These resulted in pUC18T-miniTn7T-Tc-lacIq-Ptac::sodB and pUC18T-miniTn7T-Tc-lacIq-Ptac::sodC, respectively.

To construct the AB5075 derivatives bearing *PsodB* and *PsodC* promoter fusions to *gfpmut3*, a pUC18T-miniTn7T-zeo-gfpmut3 vector was used as backbone for cloning[76]. A 1 kb DNA fragment including the *PsodB* promoter region was PCR amplified using primers sodB trx fw EcoRI abd sodB trx rv BamHI, digested with EcoRI and BamHI and cloned into pUC18T-miniTn7T-zeo-gfpmut3 cut with the same enzymes. Similarly, A 1 kb DNA fragment including the *PsodC* promoter region was PCR amplified using primers sodC trx fw EcoRI and sodC trx rv BamHI, digested with EcoRI and BamHI and ligated into pUC18T-miniTn7T-zeo-gfpmut3 cut with the same enzymes. These resulted in plasmids pUC18T-miniTn7T-zeo-PsodB::gfpmut3 and pUC18T-miniTn7T-zeo-PsodC::gfpmut3, respectively.

To integrate miniTn7-based constructions in the *att*Tn7 neutral chromosomal site, a previously established four-parental mating strategy was followed[77], using pRK2013 and pTNS2 as helper plasmids[76,78]. Selection was performed on LB agar supplemented with gentamycin (50 mg/L) and either tetracycline (2.5 mg/L) or zeocin (300 mg/L), depending on the selection marker carried in the miniTn7T backbone. Insertions were validated by PCR using oligonucleotides AB5075-glmS fw and Tn7R (de Dios et al.[75]).

**Phytochemical potentiator screen**. *A. baumannii* overnight cultures were diluted in cation-adjusted Mueller-Hinton broth

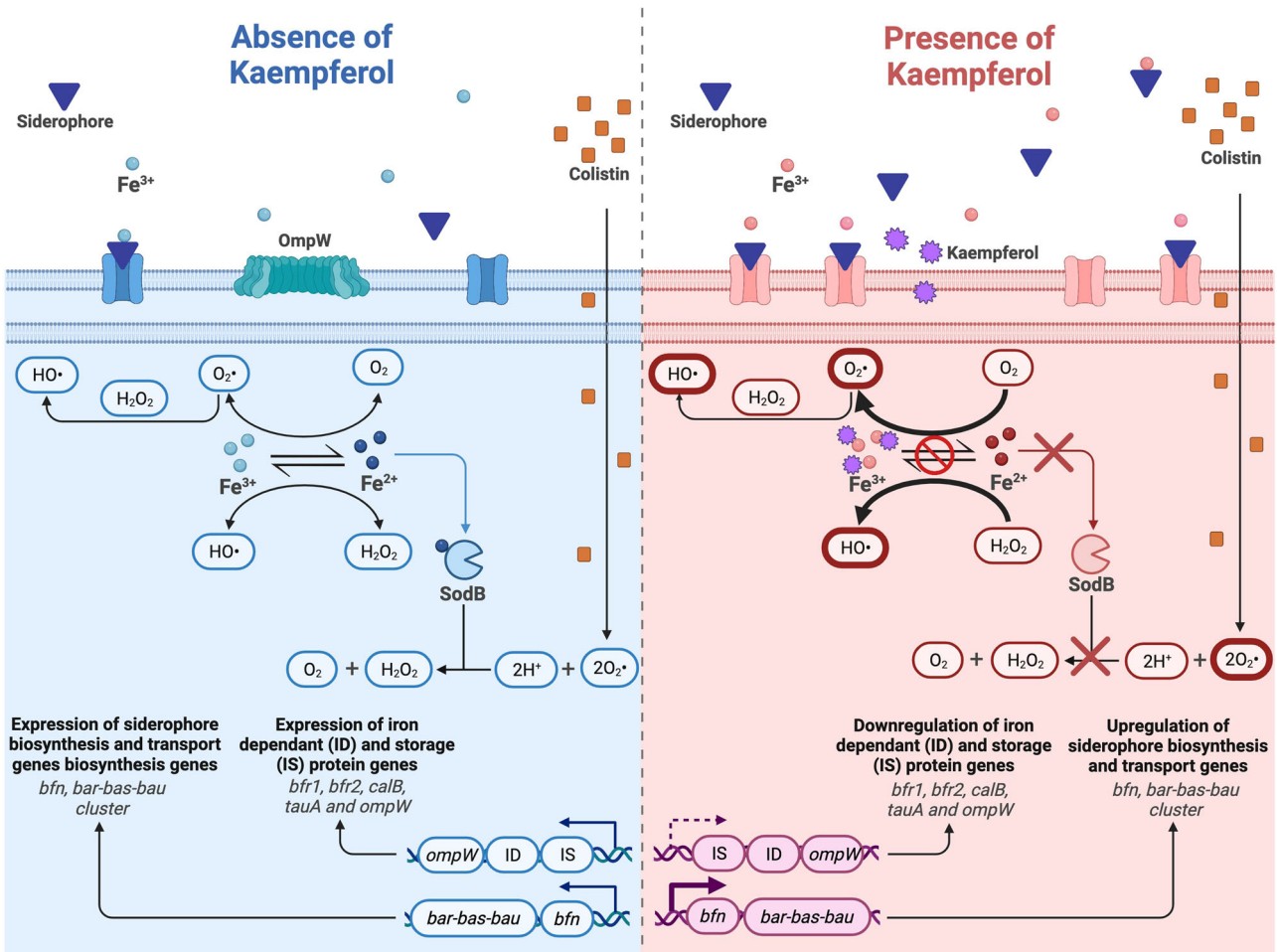

**Fig. 10 Model for the mechanism of synergy between kaempferol and colistin.** The blue section of the diagram (left) describes cellular processes when bacteria are exposed only to sub-MIC amounts of colistin. In this case, ROS production can be reversed as Fenton's reaction can take place, while the superoxide generated from the action of colistin can be converted to less toxic products by SodB. A normal expression of iron-dependant and iron-storage proteins, as well as siderophore biosynthesis and transport genes is depicted. The red section of the diagram (right) describes the dysregulation of iron homeostasis caused by the action of kaempferol that chelates $Fe^{3+}$, including downregulation of genes encoding iron-dependent and iron-storage proteins and upregulation of siderophore biosynthesis and transport genes. Dysregulation of iron homeostasis, in turn, results in accumulation of ROS due to the action of colistin and inhibition of the Fenton reaction (thick red borders). In addition, imbalance in the intracellular iron content makes less iron available for SodB, and therefore inhibits the superoxide de-toxification process. Eventually this accumulation of ROS leads to extensive damage to lipids, proteins, nucleic acids and, ultimately, cell death. The figure was created using BioRender.com.

(pH 7.4, CAMHB) to $OD_{600}$ 0.1. A sub-MIC of colistin (1.22 µg/ml) was added to the bacterial suspension. 199 µl of the suspension was added to each well on a 96-well plate followed by the addition of 1 µl (0.05 mM) of each phytochemical from a stock concentration of 10 mM (Caithness Library). In total, 1 µl of DMSO was added as a vehicle control. In parallel, these assays were conducted in the presence of a sub-MIC of colistin (1.22 µg/ml). In this case, a growth control of the bacterial suspension supplemented with the sub-MIC of colistin was included. $OD_{600}$ readings were taken every 10 min over 12 h with a Clariostar Plus plate reader (BMG LabTech), at 37 °C, 200 rpm. All hits (compounds that produced a significant reduction in growth in the presence of a sub-MIC of colistin compared to the absence of colistin) were validated by repeating this assay with just those phytochemicals for a further three biological repeats. The results represent the average of three biological replicates ± S.D.

**Biofilm assay**. For screening antibiofilm activity, we used the same experimental setup as for the antibiotic/potentiator screening, using LB media instead of CAMBH. The plates were

then incubated at 37 °C, 200 rpm, for 16 h. Once grown, the biofilm was stained using the crystal violet method[79], with mild modifications (washes were performed by pipetting and the staining was performed with 1% crystal violet). The absorbance of ethanol solubilised crystal violet was then read at 600 nm using a Clariostar Plus plate reader (BMG LabTech). All hits (compounds that produced a significant reduction in biofilm formation with respect to the vehicle control) were validated by repeating this assay with just those phytochemicals for a further three biological repeats. The results represent the average of three biological replicates ± S.D.

**Minimum inhibitory concentration (MIC) and minimum biofilm inhibitory concentration (MBIC) assays**. *A. baumannii* overnight cultures were diluted in CAMHB (pH 7.4) to $OD_{600}$ 0.1. A twofold sub-MIC of colistin was added to the initial suspension. 200 µl of the bacterial suspension were added to a 96-well plate. To the first well 1 µl (0.05 mM) of kaempferol, from a 10 mM stock, was added. The following wells had an increased volume of kaempferol added by 0.5 µl each time. As a control,

200 μl of the bacterial solution and 200 μl of sterile CAMBH was plated in separate wells and tested. A control assay was performed in parallel using equivalent volumes of DMSO as a vehicle control. The 96-well plate was then incubated at 37 °C, 200 rpm. Endpoint $OD_{600}$ was measured after 16 h using a Clariostar Plus plate reader (BMG LabTech). MIC was defined as the lowest kaempferol concentration that completely inhibited bacterial growth in the presence of sub-MIC amounts of colistin (Supporting Information Fig. S6). The results represent the average of three biological replicates ± S.D. For MBIC testing, the same experimental setup as for MIC assessment was followed in the absence of colistin. Plates were incubated for 16 h at 37 °C, 200 rpm. Subsequently, biofilms were stained using the crystal violet protocol as explained above. MBIC was determined as the lowest kaempferol concentration that completely inhibited biofilm formation. The results represent the average of three biological replicates ± S.D. When evaluating colistin MICs for colistin resistant *E. coli* and *A. baumannii* strains, the assays were carried out in accordance with the EUCAST recommendations using broth microdilution. A series of the following colistin (MP Biomedicals) concentrations was prepared individually: 256 μg/mL, 128 μg/mL, 64 μg/mL, 32 μg/mL, 16 μg/mL, 8 μg/mL, 4 μg/mL, 2 μg/mL, 1 μg/mL, 0.5 μg/mL, 0.25 μg/mL in $CaCl_2$ (0.223 mM) supplemented MH broth and transferred to a clear-bottomed 96-well microtiter plate. Either DMSO (carrier control) or kaempferol (final concentration of 50 μM) were added to the medium, as required. IPTG (0.5 mM) was added to induce *mcr* expression for the *E. coli* K-12 strain carrying pDM1-mcr-1. Overnight cultures of each strain were standardised and added to the wells at approximately $1 × 10^5$ colony-forming units (CFU) per well and the plates were incubated for 18–24 h at 37 °C. The MIC was defined as the lowest antibiotic concentration with no visible bacterial growth in the wells. The results represent the average of three biological replicates ± S.D.

**Growth assays**. Overnight *A. baumannii* AB5075 cultures were diluted in LB broth to get an $OD_{600}$ of 0.1. Treatments consisted of kaempferol only (0.375 mM), the sub-MIC of colistin (1.22 μg/ml) and the combination treatment of kaempferol and the sub-MIC of colistin, using the respective DMSO vehicle controls. The plate was then incubated at 37 °C, 200 rpm in a Clariostar Plus plate reader (BMG LabTech), where $OD_{600}$ readings were taken every 10 min for 12 h. When testing if the overexpression of *sodB* or *sodC* affected the efficacy of the combination treatment the same experimental design was followed. The respective AB5075 derivative strains bearing the overexpression insertions (AB5075/miniTn7T-Tc-lacIq-Ptac::sodB and AB5075/miniTn7T-Tc-lacIq-Ptac::sodC) or the empty control (AB5075/miniTn7T-Tc-lacIq-Ptac) were used. When indicated, LB broth was supplemented with IPTG 1 mM to induce expression from the *Ptac* promoter. The results represent the average of three technical replicates and three biological replicates ± S.D.

**RNA-seq and gene set enrichment analysis (GSEA)**. AB5075 cells were grown in 20 ml CAMHB (pH 7.4) to mid-exponential phase ($OD_{600}$ 0.60.7) in either the presence of kaempferol or DMSO. To preserve RNA integrity, the bacterial cells were then centrifuged and washed in RNAlater. The RNA was then isolated using the RNAeasy Kit with in-column DNAase digestion (Qiagen). The RNA integrity of each sample was determined using a Bioanalyzer (Agilent 2100 Bioanalyzer and Agilent RNA 6000 Nano Kit), according to the amplitude and sharpness of the peaks corresponding to the 23 S and 16 S rRNAs. Sequencing and downstream analyses were performed at Microbial Genome Sequencing Centre (Pittsburgh, Pennsylvania, U.S.A), using an Illumina MiSeq,

with 12 million reads per sample. Quality control and adapter trimming was performed with bclfastq. Read mapping was performed with HISAT. Differential expression analysis was performed using edgeR's exact test for differences between two groups of negative-binomial counts with an estimated dispersion value of 0.1, using the *A. baumannii* AB5075-UW genome annotation as reference[49]. The volcano plot was generated using R, by plotting the log fold change on the *x* axis and p value on the *y* axis. 117 genes were differentially expressed based on a log fold change ≥1 and *p* value < 0.05 (adjusted *p* value). Peak profiles were used in order to determine RNA integrity. A Gene Set Enrichment Analysis (GSEA) was performed using FUNAGE-Pro with the default parameters[50].

**Rescue assay**. *A. baumannii* AB5075 overnight cultures were diluted in LB to an $OD_{600}$ of 0.1. Treatments were set up in a 96-well plate and consisted of kaempferol (0.375 mM), the sub-MIC of colistin (1.22 μg/ml) and the combination treatment of kaempferol and the sub-MIC of colistin, with the respective DMSO vehicle controls. The media was supplemented with 100 μM $FeCl_3$, $FeCl_2$, $CuCl_2$, $MgCl_2$, $ZnCl_2$ or $CaCl_2$. The plate was then incubated at 37 °C, 200 rpm, in a Clariostar Plus plate reader (BMG LabTech), where a $OD_{600}$ reading was taken every 10 min for 12 h. The results represent the average of three technical replicates and three biological replicates ± S.D.

**Quantifying intracellular iron**. AB5075 overnight cultures were diluted in LB to an $OD_{600}$ of 0.1. Samples were then treated and incubated for 2 h at 37 °C, under aerobic conditions. Treatments consisted of kaempferol (0.375 mM), colistin (1.22 μg/mL), combination of colistin and kaempferol and DMSO carrier control. Following the incubation period, the samples were centrifuged and resuspended in 200 μL of PBS. The samples were then sonicated on ice. 100 μL of each sample was put into a well on a 96-well plate. For the $Fe^{2+}$ assay, 5 μL of the assay buffer was added to each sample, for the $Fe^{3+}$ assay, 5 μL of the iron reducer was added. The plate was then incubated at 37 °C for 30 min. 100 μL of the iron probe was then added and incubated under those same conditions for 1 h. The plate was then measured on a colorimetric plate reader (OD 593 nm), Clariostar Plus (BMG LabTech). Standards were carried out according to the iron kit (Abcam, iron kit) protocol.

**NPN uptake assay**. Mid-log phase cultures of *A. baumannii* were diluted to $OD_{600}$ 0.5 in 5 mM HEPES (pH 7.2) and 100 μL was transferred to clear-bottomed 96-well microtiter plates (Corning). Kaempferol was added to the final concentration of 0.375 mM and colistin sulphate (in 5 mM HEPES, Thermo Scientific) was added to a final concentration of 1.22 μg/mL (for experimental conditions) or 16 μg/mL (for positive control). Equal volumes of the kaempferol carrier (DMSO) was included in colistin only wells as required. 1-N-phenylnaphthylamine (NPN) (Acros Organics) was then added to a final concentration of 10 μM. Immediately after the addition of NPN, fluorescence was measured at 1-minute intervals for 20 min using a Synergy H1 microplate reader (BioTek); the excitation wavelength was set to 355 nm and emission was recorded at 405 nm[80].

**PI uptake assay**. Mid-log phase cultures *A. baumannii* were centrifuged, resuspended in phosphate buffered saline (PBS, pH 7.4), diluted to a final $OD_{600}$ of 0.4 and 100 μL was transferred to clear-bottomed 96-well microtiter plates (Corning). Kaempferol was added to the final concentration of 0.375 mM and colistin sulphate (in PBS, Thermo Scientific) was added to a final concentration of 1.22 μg/mL (for experimental conditions) or 16 μg/mL (for positive control). Equal volumes of the kaempferol

carrier (DMSO) was also included in the colistin only wells as required. Propidium iodide (PI, Acros Organics) was then added at a final concentration of 3 μM and the plate was incubated at room temperature for 10 min. The PI fluorescence was measured at 1-minute intervals for 20 min using a Synergy H1 microplate reader (BioTek); the excitation wavelength was set to 493 nm and emission was recorded at 636 nm.

**ROS production**. *A. baumannii* AB5075 overnight cultures were diluted in LB 1/100 (v/v) and grown to an $OD_{600}$ of 0.5 at 37 °C, 200 rpm. Cells were then washed three times with PBS and resuspended in PBS. Cells were then treated with 20 μM 2′,7′-dichlorofluorescin diacetate (DCFDA) and incubated at 37 °C for 2 h in the dark. Following this incubation period, samples were placed in a 96-well plate. Fluorescence intensity (excitation: 488 nm; emission: 530 nm) was measured in a Clariostar Plus plate reader (BMG LabTech). The fluorescence readings were then normalised against the $OD_{600}$ readings of the samples. The results represent the average of three technical replicates and three biological replicates ± S.D.

**GFP-based expression assay**. The expression from the *PsodB* and *PsodC* promoters was measured using miniTn7T insertions carrying either a *PsodB::gfpmut3* or a *PsodC::gfpmut3* transcriptional fusion (strains AB5075/miniTn7T-zeo-PsodB::gfpmut3 and AB5075/miniTn7T-zeo-PsodC::gfpmut3, respectively). An AB5075/miniTn7T-zeo-gfpmut3 strain (carrying an empty control), was used as a baseline control. Saturated overnight cultures of the different strains were diluted 1:100 (v/v) in fresh LB broth supplemented with the kaempferol and colistin combination treatment or a DMSO mock treatment. Cultures were incubated for 2 h at 37 °C, 180 rpm. Afterwards, 1 mL samples were washed with PBS and eventually resuspended in PBS. Then, samples were placed in a 96-well plate and their $OD_{600}$ and GFP fluorescence (excitation: 485 nm; emission: 535 nm) were measured in a Clariostar Plus plate reader (BMG LabTech). The fluorescence readings were normalised by their respective $OD_{600}$ and the baseline fluorescence obtained from the empty transposon control was subtracted from those obtained with the strains bearing either the *PsodB::gfpmut3* or the *PsodC::gfpmut3* the promoter fusions. Three biological replicates (two technical replicates each) were performed for each experimental condition.

**Cytotoxicity and in vivo efficacy**. All in vivo experiments were conducted using *G. mellonella* (Live Foods Ltd.) of 0.278 g average weight. *A. baumannii* AB5075 overnight cultures were diluted in PBS to an $OD_{600}$ of 1.0 and then further diluted 1/1000 (v/v) in PBS, to get a concentration of approximately $2.56 \times 10^5$ CFU/ml. 10 μl of each diluted bacterial solution was then injected into the *G. mellonella* larvae. Following a 15-min incubation period at room temperature, the *G. mellonella* were then injected with 10 μl of kaempferol (0.5 mM), an in vivo sub-MIC of colistin (0.8 μg/*Galleria*) and the combination treatment of kaempferol and the in vivo sub-MIC of colistin, compared to the respective DMSO vehicle controls. Larvae survival was assessed over 32 h and complete melanisation with no response to mechanical stimulation was recorded as a death. A total of 30 larvae were tested per condition, performing 10-larvae repeats on different days. Prior to these assays, the same method was followed to assess the in vivo sub-MIC of colistin in *G. mellonella*. Cytotoxicity controls were carried out using the exact same methodology but injecting the larvae with PBS instead of *A. baumannii* AB5075. In vivo controls and MIC data can be found in the Supporting Information (Supporting Information Figs. S4 and S5).

**Statistics and reproducibility**. All assays were carried out in biological triplicate ($n = 3$), with three technical repeats, individual data points are included in each graph. Analysis consisted of ANOVA tests comparing the treated samples with the respective carrier control. Average values ± S. D. are represented. Significance is indicated as ns = non-significant, $*p \leq 0.05$, $**p \leq 0.01$, $***p \leq 0.001$, $****p \leq 0.0001$. All data is available upon reasonable request. Additional methods are available in the Supporting Information Methods.

**Reporting summary**. Further information on research design is available in the Nature Portfolio Reporting Summary linked to this article.

## Data availability

Transcriptomics datasets obtained through this work have been deposited at the Gene Expression Omnibus (GEO) repository of National Centre for Biotechnology Information (NCBI) under the accession number GSE212989. Numerical source data is available in the Supplementary Data 3 file and all other data are available from the corresponding author on reasonable request.

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

## Acknowledgements

R.R.M.C. and R.D. are supported by a BBSRC New Investigator Award BB/V007823/1. R.R.M.C. is also supported by the Academy of Medical Sciences/the Wellcome Trust/ the Government Department of Business, Energy and Industrial Strategy/the British Heart Foundation/Diabetes UK Springboard Award [SBF006\1040]. D.A.I.M. was supported by the National Institute of Allergy and Infectious Diseases of the National Institutes of Health under Award Number R01AI158753; the content of this study is solely the responsibility of the authors and does not necessarily represent the official views of the National Institutes of Health. We thank Dr Chris Proctor for discussions on assay development and comments on our manuscript. We also thank Christopher D'Souza for help with dRNA-Seq analysis.

## Author contributions

K.G.: Data curation; formal analysis; investigation; methodology; writing—original draft, writing—review and editing. R.D.D.; Data curation; formal analysis; investigation; methodology; writing—original draft, writing—review and editing. N.K.: Data curation; formal analysis; investigation; methodology; writing—original draft, writing—review and editing. T.A.K.P.: investigation; methodology; writing—original draft, writing—review and editing. D.A.I.M.: investigation; supervision; methodology; writing—original draft, writing—review and editing. R.R.M.C.: Conceptualization; data curation; formal analysis; supervision; funding acquisition; investigation; writing—original draft; project administration; writing—review and editing.

## Competing interests

The authors declare no competing interests.
