## [Peer Review File · Communications Biology]

Reviewers' comments:

Reviewer #1 (Remarks to the Author):

In this study, Gadar and colleagues investigate the potentiation of colistin against *Acinetobacter baumannii*, a clinically significant pathogen. From an initial screen of phytochemicals, they identified kaempferol as a colistin potentiator that was able to prevent biofilm formation on its own, but lacked intrinsic antibiotic activity against planktonic *A. baumannii*. RNA-seq of *A. baumannii* treated with kaempferol revealed 117 differentially expressed genes from untreated cells, many of which were related to iron homeostasis. The authors conducted proof-of-concept experiments to connect the decreased survival against colistin plus kaempferol to the combination of dysregulated iron homeostasis and increased reactive oxygen species (ROS). Finally, the authors demonstrate the potential of this combination therapy to resensitize colistin-resistant *A. baumannii* or treat infection *in vivo*, based off its efficacy in a *Galleria mellonella* infection model.

Overall, this manuscript follows a clear line of logic, provides proper controls for experiments, presents reproducible data across multiple biological and technical replicates, and supports an exciting therapeutic combination against *A. baumannii*, along with a model proposing the underlying mechanism. However, a shortcoming of this manuscript is the omission of reference to a recent publication from a different research group (Zhou et al., PMID: 36314964) which describes the potentiation of colistin by kaempferol in different species, including *A. baumannii*. Although both papers include similar experiments (growth inhibition assays, biofilm inhibition assays, *G. mellonella* infection models), this manuscript by Gadar et al. distinguishes itself by looking into the mechanism of potentiation.

I suggest the following modifications to the manuscript:

1. Address discrepancies in the findings of the current manuscript and those of Zhou et al. I recommend conducting experiments using the same concentrations of colistin and kaempferol as Zhou et al. to (potentially) resolve discrepancies in the growth inhibition data (Zhou et al. found that 8ug/mL kaempferol with 1ug/mL colistin was inhibitory for 2 out of 6 of their *A. baumannii* strains; 8ug/mL kaempferol is about 10-fold lower concentration than used in this manuscript) and biofilm growth inhibition data (Zhou et al. found that kaempferol alone was insufficient to prevent biofilm formation in their strains of *A. baumannii*). At minimum, the authors should discuss possible reasons for the observed discrepancies and cite Zhou et al. in that discussion.

2. Lines 310-311: The authors state that, "We have shown that in *A. baumannii* kaempferol sequesters Fe³⁺ ions (Figures 4 and 5)..." This claim is not directly supported by the data, as no structural/biochemical assays were done to show sequestration. I recommend quantifying intracellular ferric/ferrous iron fractions in the presence or absence of kaempferol to draw a more direct conclusion. Furthermore, to support the proposed model that kaempferol sequesters ferric iron and thus decreases SodB function, I would suggest quantifying iron-bound and iron-free SodB in the presence/absence of kaempferol.

3. Lines 320-322: The authors' decision to use colistin-resistant strains with defined lipid A modifications is scientifically sound, but to be more comprehensive, I suggest testing kaempferol in a colistin-resistant strain(s) that lacks lipid A modifications as the mechanism of resistance. For example, would the authors expect kaempferol to potentiate colistin against a strain that is resistant due to increased efflux pump expression? This would have important implications for how broadly this combined therapy can be applied.

4. Lines 359-360: Maslova et al. does not seem like an appropriate citation for the *G. mellonella* infection model used in this manuscript because the former describes a biofilm and burn model that, to my understanding, the authors did not use for this paper. A different review by Ménard et al. seems more appropriate (PMID: 35004350).

5. Line 379: I recommend changing the data presentation to match Figure S1 where the x-axis starts at 24 hours, not 23 hours. It doesn't appear that data were taken at 23 hours, so the current graph is portraying a false "drop" in survival over a mere 1 hour period.

6. Line 577: As written, the stated p-value cut-off of < 0.05 for identifying differentially expressed genes would give an inappropriately high false discovery rate. Based on their use of edgeR, I'm inferring that 0.05 is the adjusted p-value, but the authors should clarify their statistical methods more precisely here. Additionally, in lines 567-568, the statement, "The RNA integrity of each sample was determined using a bioanalyzer," is not particularly informative. The authors should state the metrics used for RNA integrity and/or the model of bioanalyzer that was used.

7. Line 609: Provide detail on how the *G. mellonella* survival was determined beyond stating, "melanisation and response to mechanical stimulation." Was a scoring system used, or were the methods more "all or nothing" similar to those in McCarthy et al. 2017 (based on lack of movement upon stimulation and darkened color)? Ideally, the supplemental data would include images of the replicates.

8. Line 612: For cytotoxicity testing, mention that negative controls were injected with PBS in the Methods section. As written, it sounds like the negative control was a non-injection control.

Technical Edits:

1. Line 62: Change "1950s" to "1950's"
2. Line 320: Missing a period after (Dortet et al., 2018)
3. Line 371: "threatened" should be changed to "treated"
4. Line 452: On the lower right side of the figure, "Upregulation of siderophore biosynthesis and transport genes" is obscuring text beneath it
5. Line 452: Enlarge the font size for the iron ions in the figure, as the superscript is too small to read as is
6. Line 466: Have the authors provided documentation showing that they have permission to publish the figure created on BioRender?
7. Line 530: Missing period after "added"
8. Line 604: Missing space and period between "CFU/mL" and "10"
9. Line 607: Missing space between "Galleria)" and "and"
10. Line 610: Missing space between "days." and "Prior"
11. Lines 219, 421, 435, and 681: Refer to the reference by Dimitric Markovic et al. consistently throughout the manuscript, as sometimes it is written as "Dimitric Markovic" and other times as just "Markovic," which makes the source difficult to find in the reference list.
12. Spell check for *A. baumannii*: there are a few missing n's in the supplement (strains table) and one in the main text.

Reviewer #2 (Remarks to the Author):

Review for "Targeting iron homeostasis as a means to potentiate colistin treatment"

Brief Summary:

The overall goal of the study is to characterize the mechanism of action using a combinatorial treatment of sub-inhibitory colistin concentrations and a phytochemical called kaempferol. While colistin is available as a last line therapeutic, it is very toxic and oftentimes causes nephrotoxicity in patients. Using lower colistin concentrations could be beneficial to improve treatment outcomes. Here, the authors screen for phytochemicals to potentiate *Acinetobacter baumannii* (Ab) killing in combination with sub-inhibitory colistin concentrations. They found that kaempferol is somewhat potent in combination with colistin. Alone, kaempferol is not active against Ab, but does inhibit biofilm formation, along with several derivatives. It was known that kaempferol chelates iron ions, and here the authors show that it also disrupts iron homeostasis in Ab.

Overall thoughts:

While the manuscript is straightforward and interesting, similar antimicrobial combination studies have been previously reported using another Gram-negative pathogen, *Klebsiella pneumoniae* (PMID: 36314964). In that report, the studies also showed growth inhibition of Ab by subinhibitory colistin and kaempferol combinations. The distinguishing feature of the manuscript in review is that they characterized the antimicrobial activity of kaempferol as an iron chelator, which was also known (PMID: 9369196). Based on these previous reports, it is unclear how this report is advancing the field. It is also unclear if kaempferol is potentiating colistin-dependent killing through iron chelation and oxidative stress or if colistin is disrupting the membrane to enable kaempferol penetration into Ab, where it chelates essential iron. While the authors state that the mechanism is colistin-dependent killing through oxidative stress, their data do not support this conclusion. There are also several issues with references, reporting the methods in a reproducible manner and other mis-reported data throughout the manuscript. Therefore, this manuscript was deemed to have a very limited impact on the field.

Major concerns:

1. Similar studies in Ab and other Gram-negative pathogens have been previously reported (PMID: 36314964). However, this reference is not cited. This is a big problem, and the authors need to better distinguish how their studies advance the field relative to previous work done by other groups.
2. Many references throughout the introduction are incorrect/misplaced and fundamental studies to understand colistin resistance in Ab are not cited.
3. I disagree with the authors that their data shows that kaempferol potentiates colistin-dependent killing. It is more logical that sub-inhibitory colistin concentrations disrupt the membranes to give kaempferol access to the cytoplasm where it chelates iron to inhibit growth. The authors need to perform several additional experiments to understand kaempferol entry into Ab and how that changes when colistin perturbs the barrier.
4. There are several instances through the manuscript where data is misrepresented. For example, Figure 4 shows that 99 genes are upregulated and 18 are down regulated. The authors switched them. Mis-statements like these are common.

Minor concerns to improve clarity:

1. There are also many minor concerns, but I think that those can be addressed later after the major concerns have been addressed.

Reviewer #3 (Remarks to the Author):

The manuscript by Gadar et al., describes the effect of kaempferol on the susceptibility of *A. baumannii* and other bacteria to the antibiotic colistin. The manuscript is well written and presented and the data provide interesting and important information.

Major points:

The statistics need attention. Most graphs have multiple data points, which make them unsuitable for T-tests without correction for multiple comparison. This is the case even though most analyses are pair-wise. It would be more appropriate to use one-way ANOVA (e.g. 2B), two-way ANOVA (e.g. 1A) or two-way repeated measures ANOVA (1B). An alternative for Figure 1B would be to measure the area under the curve. All of these tests can be done using e.g. Graphpad Prism software. As a minor additional point here, it's not clear what the bars are doing between the graph and the asterisk in e.g. 2B or 3A.

There's not enough information provided on the strains used. A table detailing drug resistance profiles and the type of LPS modification involved would be very useful.

The main conclusion from the work is that iron chelation is the key mechanism by which kaempferol potentiates colistin. The work with adding back iron is interesting, but it's a high concentration and it's not clear why this was chosen – how much iron does kaempferol chelate and theoretically how much free iron would there be under these conditions? The use of EDTA is not ideal because it chelates many different cations, including Ca and Mg that stabilise the outer membrane of Gram-negative bacteria. I would like to see a more specific iron chelator used here or use something like chelex to remove all the metals and then show specificity with a justified concentration of each metal.

The statement on line 310-311 claiming to show kaempferol sequesters Fe³⁺ is incorrect. There is no direct evidence of chelation.

For the ROS assay, it's not clear what ROS specifies this dye detects? These dyes can be non-specific so caution should be applied here (e.g. PMID: 25666086). Do the authors have additional evidence for oxidative stress here such as upregulation of genes related to oxidative stress resistance? Further, since the concentration of H₂O₂ used as a control here gave similar fluorescence to colistin + kaempferol, was this concentration sufficient to inhibit growth?

Minor points

Figure 9. The diagram should have two membranes. It would also benefit from some indication of how colistin is proposed to trigger ROS production.

In several places, the results use the present tense.

Red - Reviewers comments

Black - Response

Reviewer #1 (Remarks to the Author):

In this study, Gadar and colleagues investigate the potentiation of colistin against *Acinetobacter baumannii*, a clinically significant pathogen. From an initial screen of phytochemicals, they identified kaempferol as a colistin potentiator that was able to prevent biofilm formation on its own, but lacked intrinsic antibiotic activity against planktonic *A. baumannii*. RNA-seq of *A. baumannii* treated with kaempferol revealed 117 differentially expressed genes from untreated cells, many of which were related to iron homeostasis. The authors conducted proof-of-concept experiments to connect the decreased survival against colistin plus kaempferol to the combination of dysregulated iron homeostasis and increased reactive oxygen species (ROS). Finally, the authors demonstrate the potential of this combination therapy to resensitize colistin-resistant *A. baumannii* or treat infection in vivo, based off its efficacy in a *Galleria mellonella* infection model.

Overall, this manuscript follows a clear line of logic, provides proper controls for experiments, presents reproducible data across multiple biological and technical replicates, and supports an exciting therapeutic combination against *A. baumannii*, along with a model proposing the underlying mechanism. However, a shortcoming of this manuscript is the omission of reference to a recent publication from a different research group (Zhou et al., PMID: 36314964) which describes the potentiation of colistin by kaempferol in different species, including *A. baumannii*. Although both papers include similar experiments (growth inhibition assays, biofilm inhibition assays, *G. mellonella* infection models), this manuscript by Gadar et al. distinguishes itself by looking into the mechanism of potentiation.

I suggest the following modifications to the manuscript:

1. Address discrepancies in the findings of the current manuscript and those of Zhou et al. I recommend conducting experiments using the same concentrations of colistin and kaempferol as Zhou et al. to (potentially) resolve discrepancies in the growth inhibition data (Zhou et al. found that 8ug/mL kaempferol with 1ug/mL colistin was inhibitory for 2 out of 6 of their *A. baumannii* strains; 8ug/mL kaempferol is about 10-fold lower concentration than used in this manuscript) and biofilm growth inhibition data (Zhou et al. found that kaempferol alone was insufficient to prevent biofilm formation in their strains of *A. baumannii*). At minimum, the authors should discuss possible reasons for the observed discrepancies and cite Zhou et al. in that discussion.

We sincerely thank the reviewer for their insightful and constructive comments on our manuscript. We appreciate their suggestion to address discrepancies in our findings and those of Zhou *et al.* regarding the effect of kaempferol and colistin on *A. baumannii* strains. As they rightly pointed out, Zhou *et al.* reported different levels of susceptibility for the strains they used, and the amount of kaempferol/colistin required to limit growth varied between strains. Our study, which is mostly focused on the mechanism of action of kaempferol, is largely based on a widely available and well characterised multidrug-resistant clinical isolate of *A. baumannii*, strain AB5075. Strain-specific differences between this strain and the ones used by Zhou *et al.*, are likely to account for the differences in the efficacy of the kaempferol/colistin combination treatment, something that is not surprising when comparing antibiotic susceptibility between diverse clinical isolates. That said, we do profile the full range of activity of the kaempferol/colistin combination treatment on *A. baumannii* AB5075 in Figure 1A of our study, allowing the readers to compare our findings with the ones from Zhou *et al.* 2022.

With respect to biofilm formation, our results demonstrate that kaempferol at a concentration of 0.05 mM has a relatively minimal effect on biofilm (Figure 3B). Therefore, lower concentrations, such as those used in the Zhou *et al.* study, are unlikely to anti-biofilm activity, corroborating their findings. Nonetheless, in our study we test a range of higher kaempferol concentrations, and show that 0.3 mM reduces biofilm by >90% compared to the carrier control, critically without impacting bacterial growth (Figure 3B). This observation further distinguishes our work from the Zhou *et al.* study, and highlights the promising anti-biofilm activity of kaempferol and its derivative compounds. Lines: 193-197.

2. Lines 310-311: The authors state that, “We have shown that in *A. baumannii* kaempferol sequesters Fe³⁺ ions (Figures 4 and 5)...” This claim is not directly supported by the data, as no structural/biochemical assays were done to show sequestration. I recommend quantifying intracellular ferric/ferrous iron fractions in the presence or absence of kaempferol to draw a more direct conclusion. Furthermore, to support the proposed model that kaempferol sequesters ferric iron and thus decreases SodB function, I would suggest quantifying iron-bound and iron-free SodB in the presence/absence of kaempferol.

We thank the reviewer for their insightful and valuable feedback. We have taken their comments into consideration and have performed several additional assays aiming to address this concern. Considering the significant effort required to quantify iron-bound and iron-free SodB in the presence and absence of kaempferol, we chose to provide data regarding the role of kaempferol in affecting SodB protein function through several other experimental avenues. First, we have overexpressed SodB and SodC separately in a wild-type *A. baumannii* AB5075 background and assessed the lethality of our colistin/kaempferol treatment on these strains. Here, we reasoned that if iron is essential for SodB activity, then overexpressing *sodB* in the presence of the combination treatment should not rescue the cells since this enzyme would still be iron depleted. We also hypothesized that in these conditions, overexpression of *sodC*, which is not dependent on iron availability should provide some benefit. Indeed, our assays confirmed that this is the case with only *sodC* overexpression resulting in a partial rescue phenotype (Figure 7C). Considering that overexpression of *sodC* only partially rescues the lethality of the combination treatment, we posited that SodB has a more important role in ROS detoxification than SodC. Further analysis of our RNASeq data, revealed that *sodB* and *sodC* are expressed at different levels, with *sodB* being expressed at a higher level than *sodC*. We validated this by generating *gfp* promoter fusions of both *sodB* and *sodC*, which confirmed the RNA-Seq observation and showed that *sodB* is expressed at a higher level than *sodC* at physiological conditions (Figure 7D). We then used these reporters to assess the transcription of these genes during combination treatment and recorded no difference in transcription levels under stress. These results, together with our *sodB* and *sodC* overexpression assays, support the claim that the iron-chelating capacity of kaempferol inhibits the enzymatic function of SodB (Figure 7D). Lines: 361-389.

As the reviewer suggested, to support our claim that kaempferol sequesters Fe³⁺ ions in *A. baumannii*, we have quantified the levels of available intracellular Fe²⁺ and Fe³⁺ in the presence of kaempferol. Our assays demonstrate a decrease in both forms of iron in the presence of kaempferol (Fig. 5D). Considering the interconversion ability of the two oxidation states of iron within the cellular environment, our results indicate that only the addition of Fe³⁺ rescues the previously observed lethal effect caused by the combination of kaempferol and colistin (Figure 5A and 5B). Furthermore, our *in vitro* investigations demonstrate that kaempferol binds specifically to Fe³⁺ but not to Fe²⁺, in agreement with previous reports (Figure S1). Based on these collective observations, we propose that kaempferol functions by sequestering Fe³⁺ within the cell, and this specific binding event underpins its colistin potentiating activity.

3. Lines 320-322: The authors' decision to use colistin-resistant strains with defined lipid A modifications is scientifically sound, but to be more comprehensive, I suggest testing kaempferol in a colistin-resistant strain(s) that lacks lipid A modifications as the mechanism of resistance. For example, would the authors expect kaempferol to potentiate colistin against a strain that is resistant due to increased efflux pump expression? This would have important implications for how broadly this combined therapy can be applied.

We thank the reviewer for their suggestion and insightful feedback. Efflux-mediated colistin resistance is rare and offers a low level of resistance to colistin in comparison to other colistin resistance mechanisms (PMID: 28120193). As a result, strains with this mechanism of resistance are not readily available and are rarely encountered in the clinic. Nonetheless, we will keep this excellent suggestion in mind for future research.

4. Lines 359-360: Maslova et al. does not seem like an appropriate citation for the *G. mellonella* infection model used in this manuscript because the former describes a biofilm and burn model that, to my understanding, the authors did not use for this paper. A different review by Ménard et al. seems more appropriate (PMID: 35004350).

We have modified the text accordingly.

5. Line 379: I recommend changing the data presentation to match Figure S1 where the x-axis starts at 24 hours, not 23 hours. It doesn't appear that data were taken at 23 hours, so the current graph is portraying a false "drop" in survival over a mere 1-hour period.

We have modified the figure accordingly.

6. Line 577: As written, the stated p-value cut-off of < 0.05 for identifying differentially expressed genes would give an inappropriately high false discovery rate. Based on their use of edgeR, I'm inferring that 0.05 is the adjusted p-value, but the authors should clarify their statistical methods more precisely here. Additionally, in lines 567-568, the statement, "The RNA integrity of each sample was determined using a bioanalyzer," is not particularly informative. The authors should state the metrics used for RNA integrity and/or the model of bioanalyzer that was used.

We thank the reviewer for their insightful comments, we agree that it is important to clarify our statistical methods and the metrics used for RNA integrity. We apologize for any confusion caused by our wording. To clarify, we used the edgeR package to identify differentially expressed genes, and our stated p-value cut-off of < 0.05 refers to the adjusted p-value. We have revised the methods text to make this clearer. Regarding the RNA integrity metrics, we used peak profile to determine RNA integrity. We have now updated the text accordingly (Lines: 745-747).

7. Line 609: Provide detail on how the *G. mellonella* survival was determined beyond stating, "melanisation and response to mechanical stimulation." Was a scoring system used, or were the methods more "all or nothing" similar to those in McCarthy et al. 2017 (based on lack of movement upon stimulation and darkened color)? Ideally, the supplemental data would include images of the replicates.

We have modified the text accordingly to include the scoring system used (Lines: 835-837).

8. Line 612: For cytotoxicity testing, mention that negative controls were injected with PBS in the Methods section. As written, it sounds like the negative control was a non-injection control.

We have modified the text accordingly (Lines: 839-842).

Technical Edits:

1. Line 62: Change “1950s” to “1950’s”

We have modified the text accordingly

2. Line 320: Missing a period after (Dortet et al., 2018)

We have modified the text accordingly

3. Line 371: “threatened” should be changed to “treated”

We have modified the text accordingly

4. Line 452: On the lower right side of the figure, “Upregulation of siderophore biosynthesis and transport genes” is obscuring text beneath it

We have modified the figure accordingly

5. Line 452: Enlarge the font size for the iron ions in the figure, as the superscript is too small to read as is

We have modified the figure accordingly

6. Line 466: Have the authors provided documentation showing that they have permission to publish the figure created on BioRender?

We have modified the figure legend accordingly as the lab has a full BioRender License.

7. Line 530: Missing period after “added”

We have modified the text accordingly

8. Line 604: Missing space and period between “CFU/mL” and “10”

We have modified the text accordingly

9. Line 607: Missing space between “Galleria)” and “and”

We have modified the text accordingly

10. Line 610: Missing space between “days.” and “Prior”

We have modified the text accordingly

11. Lines 219, 421, 435, and 681: Refer to the reference by Dimitric Markovic et al. consistently throughout the manuscript, as sometimes it is written as “Dimitric Markovic” and other times as just “Markovic,” which makes the source difficult to find in the reference list.

We have modified the text and reference list accordingly

12. Spell check for *A. baumannii*: there are a few missing n's in the supplement (strains table) and one in the main text.

We have modified the text accordingly.

Reviewer #2 (Remarks to the Author):

Review for "Targeting iron homeostasis as a means to potentiate colistin treatment"

Brief Summary:

The overall goal of the study is to characterize the mechanism of action using a combinatorial treatment of sub-inhibitory colistin concentrations and a phytochemical called kaempferol. While colistin is available as a last line therapeutic, it is very toxic and oftentimes causes nephrotoxicity in patients. Using lower colistin concentrations could be beneficial to improve treatment outcomes. Here, the authors screen for phytochemicals to potentiate *Acinetobacter baumannii* (Ab) killing in combination with sub-inhibitory colistin concentrations. They found that kaempferol is somewhat potent in combination with colistin. Alone, kaempferol is not active against Ab, but does inhibit biofilm formation, along with several derivatives. It was known that kaempferol chelates iron ions, and here the authors show that it also disrupts iron homeostasis in Ab.

Overall thoughts:

While the manuscript is straightforward and interesting, similar antimicrobial combination studies have been previously reported using another Gram-negative pathogen, *Klebsiella pneumoniae* (PMID: 36314964). In that report, the studies also showed growth inhibition of Ab by subinhibitory colistin and kaempferol combinations. The distinguishing feature of the manuscript in review is that they characterized the antimicrobial activity of kaempferol as an iron chelator, which was also known (PMID: 9369196). Based on these previous reports, it is unclear how this report is advancing the field.

It is also unclear if kaempferol is potentiating colistin-dependent killing through iron chelation and oxidative stress or if colistin is disrupting the membrane to enable kaempferol penetration into Ab, where it chelates essential iron. While the authors state that the mechanism is colistin-dependent killing through oxidative stress, their data do not support this conclusion. There are also several issues with references, reporting the methods in a reproducible manner and other mis-reported data throughout the manuscript. Therefore, this manuscript was deemed to have a very limited impact on the field.

Major concerns:

1. Similar studies in Ab and other Gram-negative pathogens haven been previously reported (PMID: 36314964). However, this reference is not cited. This is a big problem, and the authors need to better distinguish how their studies advance the field relative to previous work done by other groups.

We thank the reviewer for their valuable comments. The contemporaneously produced study by Zhou *et al.*, 2022, does show the impact of kaempferol on colistin susceptibility in *A. baumannii*, however critically, it does not provide a robust mechanism of action. In our study, we have performed extensive transcriptomic, genetic and metabolic rescue experiments to confirm the mechanism of action underpinning this increased susceptibility for the first time. We have now updated the manuscript to compare the work of Zhou *et al.*, 2022 with our work, and have made a concerted effort to distinguish our study from previous work in the field throughout (Lines: 193-197, 285-300, 518-523, 598-601).

2. Many references throughout the introduction are incorrect/misplaced and fundamental studies to understand colistin resistance in *Ab* are not cited.

We have revised the references throughout the introduction and have further cited the following studies in order to better support our claims; PMID: 31765820, PMID: 21881132, PMID: 34609499, PMID: 27847502, PMID: 34661894, PMID: 2732950, PMID: 29638177, PMID: 29483899, PMID: 23230287, PMID: 25904728, PMID: 30720421, PMID: 21081544, PMID: 20006471, PMID: 22908157.

3. I disagree with the authors that their data shows that kaempferol potentiates colistin-dependent killing. It is more logical that sub-inhibitory colistin concentrations disrupt the membranes to give kaempferol access to the cytoplasm where it chelates iron to inhibit growth. The authors need to perform several additional experiments to understand kaempferol entry into *Ab* and how that changes when colistin perturbs the barrier.

We thank the reviewer for their comment and for sharing their compelling perspective. We have carefully considered their suggestion and conducted additional experiments to conclusively determine whether membrane disruption was a key part of the colistin kaempferol mechanism (something also claimed by Zhou *et al.*). We used widely accepted assays to determine inner membrane and outer membrane integrity, and we conclusively show that there is no damage to either membrane by the sub-MIC concentration of colistin used in this paper (1.22 µg/ml); as expected at higher concentrations of colistin (16 µg/ml), there is clear evidence of damage to both membrane barriers (Figure 6 Lines: 285-300). These findings demonstrate that colistin at the sub-MIC amounts used in our study does not have a role in allowing kaempferol entry to the cytoplasm, but as we show in Figure 7A does lead to drastic increase of the ROS levels. This further confirms that the inhibition of growth is not due to increased membrane permeability or disruption and supports our original hypothesis of iron homeostasis and ROS build-up being the mechanism of action. We feel that the addition of this data has greatly strengthened the conclusions of our study so again we thank the reviewer for suggesting these experiments.

4. There are several instances through the manuscript where data is misrepresented. For example, Figure 4 shows that 99 genes are upregulated and 18 are down regulated. The authors switched them. Mis-statements like these are common.

We apologise for this oversight and have now have modified the text and figures accordingly.

Minor concerns to improve clarity:

5. There are also many minor concerns, but I think that those can be addressed later after the major concerns have been addressed.

We have made a considerable effort to address all the minor concerns of the other reviewers and hope that the majority of the concerns suggested here have also been addressed with these revisions.

Reviewer #3 (Remarks to the Author):

The manuscript by Gadar et al., describes the effect of kaempferol on the susceptibility of *A. baumannii* and other bacteria to the antibiotic colistin. The manuscript is well written and presented and the data provide interesting and important information.

Major points:

1. The statistics need attention. Most graphs have multiple data points, which make them unsuitable for T-tests without correction for multiple comparison. This is the case even though most analyses are pair-wise. It would be more appropriate to use one-way ANOVA (e.g. 2B), two-way ANOVA (e.g. 1A) or two-way repeated measures ANOVA (1B). An alternative for Figure 1B would be to measure the area under the curve. All of these tests can be done using e.g. Graphpad Prism software.

We have modified the statistics accordingly throughout.

2. As a minor additional point here, it's not clear what the bars are doing between the graph and the asterisk in e.g. 2B or 3A.

We have modified the figures highlighted here to offer greater clarity.

3. There's not enough information provided on the strains used. A table detailing drug resistance profiles and the type of LPS modification involved would be very useful.

We thank the reviewer for this excellent suggestion. We appreciate their feedback and have now included a table detailing the drug resistance profiles and the type of LPS modification involved for the strains used in our study (Supporting Information Table S2).

4. The main conclusion from the work is that iron chelation is the key mechanism by which kaempferol potentiates colistin. The work with adding back iron is interesting, but it's a high concentration and it's not clear why this was chosen – how much iron does kaempferol chelate and theoretically how much free iron would there be under these conditions? The use of EDTA is not ideal because it chelates many different cations, including Ca and Mg that stabilise the outer membrane of Gram-negative bacteria. I would like to see a more specific iron chelator used here or use something like chelex to remove all the metals and then show specificity with a justified concentration of each metal.

We thank the reviewer for their insightful comments and suggestions. Regarding the use of high iron concentration, we adapted the assay from a previously published protocol and chose the dose reported in this protocol to simulate iron rescue (PMID: 20692388). However, we understand the reviewers concern and have conducted additional experiments to strengthen our conclusions. We agree that the use of EDTA may not be ideal as it chelates multiple cations, including Ca^{2+} and Mg^{2+} that contribute to the integrity of the outer membrane of Gram-

negative bacteria. Thus, we have conducted further experiments by supplementing back Ca^{2+} and Mg^{2+} into the media, and found that it had no significant impact on rescuing the growth of the treated bacteria compared to the rescue phenotype that we observe when we supplement media with Fe^{3+} . We tested other cations as well, like Cu^{2+} and Zn^{2+} , and these also could not rescue the lethality of the kaempferol/colistin treatment. Moreover, we used specific iron chelators, ExJade and 8-Hydroxyquinoline (Pierre *et al.*, 2003; Cappellini, 2007), and confirmed their ability to potentiate colistin, like kaempferol, thus further supporting our hypothesis that iron chelation is the key mechanism of action. We have included the additional results in the revised manuscript (Supporting Information Figure S2, Figure 7E and lines: 264-270, 384-389) significantly strengthening our original conclusions.

5. The statement on line 310-311 claiming to show kaempferol sequesters Fe^{3+} is incorrect. There is no direct evidence of chelation.

We appreciate the reviewer's feedback and have taken it into consideration. We have conducted additional assays to provide more evidence to support our hypothesis that kaempferol sequesters Fe^{3+} ions. Firstly, we have conducted an absorbance spectrum-based assay that demonstrates the binding of Fe^{3+} to kaempferol, highlighting the formation of a chelation complex between kaempferol and Fe^{3+} . This is in agreement to previously published research (PMID 9369196) that has also indicated kaempferol's ability to sequester Fe^{3+} ions (Supporting Information Figure S1, lines: 234-237).

To further support our claim that kaempferol sequesters Fe^{3+} ions in *A. baumannii*, we have quantified the levels of available intracellular Fe^{2+} and Fe^{3+} in the presence of kaempferol (Figure 5D). Our assays demonstrate a decrease in both forms of iron in the presence of kaempferol. Taking into account the interconversion ability of the two oxidation states of iron within the cellular environment, our results indicate that only the addition of Fe^{3+} rescues the previously observed lethal effect caused by the combination of kaempferol and colistin (Figure 5A and 5B). Furthermore, our *in vitro* investigations demonstrate that kaempferol binds specifically to Fe^{3+} but not to Fe^{2+} (Figure S1). Based on these collective observations, we propose that kaempferol functions by sequestering Fe^{3+} within the cell, and this specific binding event underpins its colistin potentiating activity.

6. For the ROS assay, it's not clear what ROS specifies this dye detects? These dyes can be non-specific so caution should be applied here (e.g. PMID: 25666086). Do the authors have additional evidence for oxidative stress here such as upregulation of genes related to oxidative stress resistance? Further, since the concentration of H_2O_2 used as a control here gave similar fluorescence to colistin + kaempferol, was this concentration sufficient to inhibit growth?

We thank the reviewer for their valuable feedback. Regarding the upregulation of oxidative stress genes, it is important to note that kaempferol alone did not trigger a ROS response, as demonstrated by the low level of fluorescence in our study (Figure 7A). Therefore, there would be no oxidative stress gene response in the RNASeq data, since the samples were only treated with kaempferol. We did test and confirm that the concentration of 10% v/v H_2O_2 used in our control was sufficient to inhibit bacterial growth as shown in Supporting Information Figure S3.

Our findings on the potentiation effect of kaempferol on colistin support the established knowledge that colistin acts via the production of ROS species (PMID: 32284036). We show that kaempferol presence in combination with colistin results in increased ROS production

compared to colistin on its own (Figure 7A). Interestingly, we also demonstrate that the function of SodB, which is one of the cellular mechanisms helping to overcome ROS production, is impaired (Figure 7C), likely due to iron chelation by kaempferol, which strips the enzyme of its cofactor. This means that the cell is even less able to detoxify the ROS produced by colistin when treated with kaempferol, which is in agreement with the increased levels of ROS observed in the kaempferol/colistin combination treatment in comparison to treatment with colistin alone (Figures 7A). Together these findings allow us to propose a model to explain the potentiation effect of kaempferol on colistin: by impairing the function of SodB, the cells' coping mechanism for the ROS produced by colistin is abrogated, allowing for this sublethal dose of colistin to have lethal consequences (Figure 7B).

Minor points:

7. Figure 9. The diagram should have two membranes. It would also benefit from some indication of how colistin is proposed to trigger ROS production.

We have modified the figure accordingly.

8. In several places, the results use the present tense.

We have modified the text accordingly.

Reviewers' comments:

Reviewer #1 (Remarks to the Author):

This revised manuscript from Gadar et al. is a marked improvement over the original draft. The authors have thoroughly addressed previous comments with appropriate experiments and discussion in the text of their manuscript. In particular, the section "Kaempferol does not increase membrane permeability" is well-structured with compelling data and discussion. Additionally, the new data on SodB/SodC and intracellular iron concentrations are compelling and strengthen the authors' proposed model for how kaempferol potentiates sub-MIC levels of colistin in *A. baumannii*, further distinguishing this manuscript from the Zhou et al. 2022 publication. I have only minor technical edits to recommend:

Main Text:

Lines 77-78: Change "Gram negative" to "Gram-negative"

Line 195: Reference to "the contemporaneous Zhou et al (2022) study" is abrupt and lacks context. Should be integrated more smoothly or simply referenced and discussed at a later point.

Line 292: NPN should be written out in full before using the abbreviation.

Line 320: "NPN" should be "PI" for this caption

Lines 321-323: Recommend moving the sentence "High amounts of colistin...permeabilize the inner membrane" to be in front of "All assays were carried out...technical repeats."

Line 506: Change "Logrank" to "Log Rank"

Line 600: Italicize "et al"

Line 601: Un-italicize "larvae"

Line 819: "1 ml" should be "1 mL"

Supporting Information:

Fig. S1, S2, S3: These figure captions are lacking titles as in Fig. S4 and S5.

Fig. S1: Authors should explicitly state the concentrations of kaempferol/iron ions that were used for the given ratios in their methods or in the figure caption. The note about replacing volumes of iron with water is appreciated, but more detail is required on the actual concentrations/volumes that were used.

Line 25: Remove the extra "." and "255" from "ns. = 255 non-significant"

Reviewer #2 (Remarks to the Author):

Review for revised (R1) version of "Targeting iron homeostasis as a means to potentiate colistin treatment"

The revised version of the manuscript is somewhat improved; however, several large issues remain. The references are still problematic. Many of the strains and assays are not well described and not standard in the field. Some of the data contradict each other making interpretation difficult. They also do not show bactericidal activity with the combination, which would be needed in the case of an infection.

Regarding impact, most of the findings from this paper were previously published in a wide range of resistant bacteria. The authors describe the work herein as focusing on the mechanism, but a different mechanism was previously shown. This was not resolved. I suspect the issue is that these authors are using a colistin resistant Ab5075 variant.

Major concerns:

1. The colistin minimal inhibitory concentration (MIC) of Ab5075 is 0.25 $\mu\text{g}/\text{ml}$. See PMID 32775156, PMID 32562384, and many others.

However, the authors report an Ab5075 MIC of $>0.5 \mu\text{g}/\text{ml}$ here, which conflicts with previously published data. I will also note that they did not report MIC values for any strains tested, so it is difficult to interpret their results. If the strain they are using is an Ab 5075 colistin resistant derivative, their data need to be interpreted differently.

Also, please describe how 1.22 $\mu\text{g}/\text{ml}$ subinhibitory colistin concentration used relates to the MIC. Is it 0.1X or 0.5X MIC?

2. The authors need to add Ab 5075 to the strain list. Original reference is PMID 2486555.

3. The authors need to state the final concentrations of phytochemicals in the M&M screen section. 1 μl of a 10 mM stock concentration is not sufficient. Additionally, for the MIC and Biofilm assays stating that 0.5 μl from a 10 mM stock is also not sufficient.

4. The authors should not rely on turbidity only to show inhibition but also report colony forming units over time. This is standard in the field.

5. There are no minimal bactericidal concentrations reported. Combinatorial treatments would not be given proactively to prevent infection (growth), antibiotics need to have bactericidal activity, which is not shown.

6. The authors need to report the titration assays from the other phytochemical screens for comparison. They show a growth assay in Fig 2B, but this reviewer is unsure why there is growth above OD 1.0 in all combinations. This is problematic and indicates the combination does not inhibit growth sufficiently to prevent infection.

Other edits:

Line 27- remove "largely relies". Colistin is a last-resort treatment, but it is not widely used. There are better/less toxic last-resort antibiotics that are preferred by clinicians.

Line 37 - replace "critical pathogen" with *A. baumannii*

Line 39 - The statement "prolonging the lifespan of colistin" is poorly worded and needs to be revised. Colistin is not alive.

Line 50 - remove "commonly" or qualify it

Line 77 - should be Gram-negative

Line 81 - needs to be rephrased

Line 88-90 - The statement does not accurately represent the publication. While some Colistin resistant strains harbor mutations in *pmrA* or *pmrB* that lead to colistin susceptibility, not all of them did. Those had insertion elements that increased *pmrC* transcription.

Lines 91-93 - The primary citations for lipid A modification here are PMID 21576434 and 21646482.

Lines 94-95 - The galactosamine reference is 23877686. I'm unsure how the Falagas reference fits. Please remove.

Line 101 - *A. baumannii* produces lipooligosaccharide, not LPS. It does not encode an O-antigen ligase. Several studies have shown the LOS product.

Line 100-103 – Please cite the primary literature here. PMID 20855724 in ATCC 19606 and PMID 27681618 in Ab 5075.

Line 185-187- what about 5-deoxy, which also reduced biofilm formation?

Line 195 - remove "contemporous" here and throughout the report. It would also be more appropriate to compare findings to the previous study in the discussion section.

Line 198 - what is a "next-generation antimicrobial". This is should be removed.

Line 221 - replace "their" with "gene expression".

Lines 286-290 should be deleted. The discussion showing differences in data from the previous report should be added to the discussion.

Line 342 -delete "as expected". It implies bias. Please only report the data in the results section.

Line 519 – delete "conducted concurrently with this study". Change a to "a previous report".

The discussion is problematic. In the first paragraph, the authors state that the mechanism they found that potentiates the combinatorial treatment was different than that previously reported, but did not offer any explanation.

The discussion also needs to include a section pointing out that the smaller Kaempferol structures potentiate colistin combinatorial treatment relative to the larger structures. They also need to discuss the membrane permeabilization found in Zhou 2022 in relation to their data.

Reviewer #3 (Remarks to the Author):

The authors have fully addressed my concerns.

Blue - Reviewers comments

Black - Response to the reviewer's comments

Reviewers' comments:

REVIEWER #1 (Remarks to the Author):

This revised manuscript from Gadar et al. is a marked improvement over the original draft. The authors have thoroughly addressed previous comments with appropriate experiments and discussion in the text of their manuscript. In particular, the section “Kaempferol does not increase membrane permeability” is well-structured with compelling data and discussion. Additionally, the new data on SodB/SodC and intracellular iron concentrations are compelling and strengthen the authors' proposed model for how kaempferol potentiates sub-MIC levels of colistin in *A. baumannii*, further distinguishing this manuscript from the Zhou et al. 2022 publication. I have only minor technical edits to recommend:

We thank the reviewer for their kind comments and for recognizing the rigor of our previous revisions to the manuscript. We have now addressed all of their minor suggestions in detail.

Main Text:

Lines 77-78: Change “Gram negative” to “Gram-negative”

We have corrected the text accordingly.

Line 195: Reference to “the contemporaneous Zhou et al (2022) study” is abrupt and lacks context. Should be integrated more smoothly or simply referenced and discussed at a later point.

We have updated the text according to this suggestion.

Line 292: NPN should be written out in full before using the abbreviation.

We have modified the text accordingly.

Line 320: “NPN” should be “PI” for this caption

We have modified the text accordingly.

Lines 321-323: Recommend moving the sentence “High amounts of colistin...permeabilize the inner membrane” to be in front of “All assays were carried out...technical repeats.”

We have modified the text according to this suggestion.

Line 506: Change “Logrank” to “Log Rank”

We have amended the text accordingly

Line 600: Italicize “et al”

We have corrected the text accordingly.

Line 601: Un-italicize “larvae”

We have amended the text accordingly.

Line 819: “1 ml” should be “1 mL”

We have amended the text accordingly.

Supporting Information:

Fig. S1, S2, S3: These figure captions are lacking titles as in Fig. S4 and S5.

We have added titles to Fig. S1,S2 and S3.

Fig. S1: Authors should explicitly state the concentrations of kaempferol/iron ions that were used for the given ratios in their methods or in the figure caption. The note about replacing volumes of iron with water is appreciated, but more detail is required on the actual concentrations/volumes that were used.

We have revised the figure legend of Fig. S1 in accordance with this suggestion. In particular, we have explicitly stated the kaempferol:iron concentrations for each ratio that was tested. Additional details on how the assay was performed are given in the first section the Supplementary Information Materials and Methods (L116-129 in the Supplementary Information file).

Line 25: Remove the extra “.” and “255” from “ns. = 255 non-significant”

We have amended the text accordingly.

REVIEWER #2 (Remarks to the Author):

Review for revised (R1) version of “Targeting iron homeostasis as a means to potentiate colistin treatment”

The revised version of the manuscript is somewhat improved; however, several large issues remain. The references are still problematic. Many of the strains and assays are not well described and not standard in the field. Some of the data contradict each other making interpretation difficult. They also do not show bactericidal activity with the combination, which would be needed in the case of an infection.

Regarding impact, most of the findings from this paper were previously published in a wide range of resistant bacteria. The authors describe the work herein as focusing on the mechanism, but a different mechanism was previously shown. This was not resolved. I suspect the issue is that these authors are using a colistin resistant Ab5075 variant.

We thank the reviewer for their comments, which we have addressed below in detail. In brief, we have now demonstrated bactericidal activity of the kaempferol-colistin treatment and

included experimental data that demonstrate that *A. baumannii* AB5075 is sensitive to colistin according to EUCAST breakpoints. We also note that *A. baumannii* AB5075, which was the main pathogen used throughout this manuscript, is the most widely used highly virulent multidrug-resistant clinical isolate of *A. baumannii*, included in over 82 published papers. As such, this strain is extremely well characterized and is considered a model for the *A. baumannii* field. Any additional clinical strains used in our study are clearly detailed with all source information available in Supporting Information Table S2.

Major concerns:

1. The colistin minimal inhibitory concentration (MIC) of Ab5075 is 0.25 µg/ml. See PMID 32775156, PMID 32562384, and many others. However, the authors report an Ab5075 MIC of >0.5 µg/ml here, which conflicts with previously published data. I will also note that they did not report MIC values for any strains tested, so it is difficult to interpret their results. If the strain they are using is an Ab 5075 colistin resistant derivative, their data need to be interpreted differently. Also, please describe how 1.22 µg/ml subinhibitory colistin concentration used relates to the MIC. Is it 0.1X or 0.5X MIC?

We appreciate the reviewer's comment regarding the reported minimal inhibitory concentration (MIC) value for colistin for *A. baumannii* AB5075. We acknowledge the existence of previous publications, such as PMID 32562384 and PMID 32775156, which both state **discrepant** colistin MIC values for this strain, and we would like to point out several additional publications that also report **different** colistin MICs (PMID: 29524882, PMID: 30662875, PMID: 30111627). Importantly, in all of these studies as well as in our manuscript, the colistin MIC values for *A. baumannii* AB5075 are below the EUCAST resistance breakpoint of *A. baumannii* that is 2 µg/mL, confirming that this strain is sensitive to colistin. To further reaffirm this point, we have now included the full colistin MIC assay in our revised manuscript (Supplementary Figure 6). It can be clearly seen from this data that 1.22 µg/mL is a sub-inhibitory concentration (~0.5x MIC), in agreement with the data we have originally presented (Figure 1A). The minor variance between the published MICs for this specific strain could be due to different laboratory protocols or the use of different sub-types of this strain, which are well-known to be in existence; we have clearly stated in our methods that we used the VIR-O sub-type.

We would also like to mention that the MIC values for all of the remaining clinical isolates used in this study are clearly presented in Figure 8. That said, to further address this comment we have now stated the MIC in text format for all clinical isolates tested in Supporting Information Table S2.

2. The authors need to add Ab 5075 to the strain list. Original reference is PMID 2486555.

We have modified the table accordingly (Supporting Information Table S2). However, to make clearer the provenance of the specific strain that we are using, we also state where we originally sourced this isolate, i.e., the Manoil Lab at University of Washington.

3. The authors need to state the final concentrations of phytochemicals in the M&M screen section. 1 µl of a 10 mM stock concentration is not sufficient. Additionally, for the MIC and Biofilm assays stating that 0.5 µl from a 10 mM stock is also not sufficient.

We have modified the text accordingly.

4. The authors should not rely on turbidity only to show inhibition but also report colony forming units over time. This is standard in the field.

Broth Microdilution is the gold-standard method for the determination of MIC values, as recommended by EUCAST and CLSI. We have adhered to these protocols that are used by clinicians and researchers worldwide and which require reads based on turbidity observed by eye and not colony forming units over time. Moreover, we demonstrate bactericidal activity of the kaempferol-colistin combination (see our reply to comment 5 below), something also reported in the published work of Zhou *et al.*, 2022 who showed that this treatment leads to growth inhibition, fully agreeing with our findings.

5. There are no minimal bactericidal concentrations reported. Combinatorial treatments would not be given proactively to prevent infection (growth), antibiotics need to have bactericidal activity, which is not shown.

We appreciate the reviewer's comment regarding the bactericidal activity of the combination. This has been described previously by Zhou *et al.*, 2022, and we have now included additional data confirming this bactericidal activity (Supplementary Figure 7). That said, we should note here that several antibiotics in clinical use are recognised as bacteriostatic, including but not limited to tetracyclines, macrolides, clindamycin, trimethoprim/sulfamethoxazole, linezolid, and chloramphenicol (PMID: 31613458).

6. The authors need to report the titration assays from the other phytochemical screens for comparison. They show a growth assay in Fig 2B, but this reviewer is unsure why is there growth to above OD 1.0 in all combinations. This is problematic and indicates the combination does not inhibit growth sufficiently to prevent infection.

Unfortunately, we cannot report the titrations from each of the other phytochemicals identified as colistin potentiators in our initial screen. These are the subject of ongoing work that will be published independently and is outside the scope and claims of the current manuscript. Describing them here would not add to the conclusions of this work and would unnecessarily compromise the novelty of our future studies.

Regarding the concern of the reviewer about the observed growth above OD 1.0 in the combinations shown in Fig. 2B for the derivatives of kaempferol, we would like to clarify that in the specific assay (Fig 2B) a concentration of only **0.05 mM** of each compound was used. As we are limited in the availability of some of these derivative compounds, this lower concentration was employed to simply determine whether any of these molecules is superior to kaempferol by assessing the initial impact of each compound on bacterial growth. To achieve complete inhibition, we would need to use a concentration of **0.375 mM** like with kaempferol (Fig. 1A). Due to amount limitations, and since none of these derivatives was found to be superior to kaempferol, we did not pursue additional inhibition experiments that would not add anything to our conclusions in this instance.

Other edits:

Line 27- remove "largely relies". Colistin is a last-resort treatment, but it is not widely used. There are better/less toxic last-resort antibiotics that are preferred by clinicians.

We have modified the text accordingly.

Line 37 – replace “critical pathogen” with *A. baumannii*

We have modified the text accordingly.

Line 39 – The statement “prolonging the lifespan of colistin” is poorly worded and needs to be revised. Colistin is not alive.

We have revised the text in accordance with this suggestion.

Line 50 – remove “commonly” or qualify it

We have modified the text accordingly.

Line 77 – should be Gram-negative

We have amended the text accordingly.

Line 81 – needs to be rephrased

We have rephrased the text accordingly.

Line 88-90 – The statement does not accurately represent the publication. While some Colistin resistant strains harbor mutations in *pmrA* or *pmrB* that lead to colistin susceptibility, not all of them did. Those had insertion elements that increased *pmrC* transcription.

We agree with the reviewer’s comment here and that is why we stated that the “The *majority* of colistin-resistant strains harbour mutations in the two-component genes *pmrA* or *pmrB*” and did not state that they all had mutations in these genes.

Lines 91-93 – The primary citations for lipid a modification here are PMID 21576434 and 21646482.

We have amended the text accordingly.

Lines 94-95 - The galactosamine reference is 23877686. I’m unsure how the Falagas reference fits. Please remove.

We have modified the reference accordingly.

Line 101 – *A. baumannii* produces lipooligosaccharide, not LPS. It does not encode an O-antigen ligase. Several studies have shown the LOS product.

We have revised the text in accordance with this suggestion.

Line 100-103 – Please cite the primary literature here. PMID 20855724 in ATCC 19606 and PMID 27681618 in Ab 5075.

We have modified the reference accordingly.

Line 185-187- what about 5-deoxy, which also reduced biofilm formation?

We appreciate the reviewers comment here. 5-Deoxykaempferol has a colistin potentiation activity and a biofilm inhibition activity very similar to that of kaempferol, likely reflecting the minor difference between their structure. We have elaborated on this in the discussion of our manuscript.

Line 195 - remove” contemporous” here and throughout the report. It would also be more appropriate to compare findings to the previous study in the discussion section.

We appreciate the reviewer's comment concerning the phrase "contemporaneous". We would like to clarify that both studies were indeed carried out simultaneously. That said, we modified the text appropriately in this instance.

Line 198 - what is a “next-generation antimicrobial”. This is should be removed.

We appreciate the reviewer's comment regarding the term "next-generation antimicrobial" (NGA) used in our paper. NGAs are compounds that may not inhibit bacterial growth on their own but possess the ability to target and inhibit processes associated with virulence and infection progression. This concept of targeting virulence factors rather than solely focusing on growth inhibition has gained recognition as a promising strategy in antimicrobial research (PMID: 24552512, PMID: 3572122). By disrupting virulence-related mechanisms, NGAs aim to attenuate the pathogenicity of bacteria and reduce the impact of infections, while reducing the chances of developing resistance. The use of the term "next-generation antimicrobial" is not unique to our study and has been widely adopted in the scientific community to describe similar approaches (PMID: 35319398, PMID: 36455341, PMID: 35675307, PMID: 36066482).

Line 221 - replace “their” with “gene expression”.

We have amended the text accordingly.

Lines 286-290 should be deleted. The discussion showing differences in data from the previous report should be added to the discussion.

We recognize the importance of directly comparing our findings to the previous study, and so have addressed this aspect in the manuscript by thoroughly discussing the similarities and differences as they arise in our analysis.

Line 342 -delete “as expected”. It implies bias. Please only report the data in the results section.

We have updated the text to reflect this suggestion.

Line 519 – delete “conducted concurrently with this study”. Change a to “a previous report”. The discussion is problematic. In the first paragraph, the authors state that the mechanism they found that potentiates the combinatorial treatment was different than that previously reported, but did not offer any explanation.

The discussion also needs to include a section pointing out that the smaller Kaempferol structures potentiate colistin combinatorial treatment relative to the larger structures. They also need to discuss the membrane permeabilization found in Zhou 2022 in relation to their data.

We appreciate the reviewer's feedback on the discussion section of our manuscript. We have carefully considered their suggestions and have made the necessary revisions to address the concerns raised. Regarding the statement about the mechanism of action being different from what was previously reported, we emphasize that our findings do not support the previously reported mechanism involving membrane permeabilization. Indeed, Reviewer 1 comments on the section of our manuscript describing membrane permeabilization and recognizes that we provide compelling data against this mode of action of the combination treatment: "In particular, the section "Kaempferol does not increase membrane permeability" is well-structured with compelling data and discussion."

We have now added additional discussion outlining that the observation of membrane damage in *A. baumannii* made by Zhou *et al.*, 2022 was based solely on alkaline phosphatase (ALPs) release at 6 hours post treatment (Lines 522-528). This is problematic as in a previous figure in the Zhou *et al.*, 2022 study (Figure 3 G and H), the published data suggests that significant bactericidal activity has occurred by 6 hours in the combination treatment, likely accounting for the observed increased levels of ALPs. Our approach using membrane specific dyes (a gold standard in the cell envelope biology field) in a much narrower time frame (less than an hour) gives a much more detailed insight into the impact of the combination treatment on the physiology of the cell; if the combination treatment was membrane permeabilizing one would expect almost instantaneous effects in our experiments.

In response to the reviewer's suggestion, we have now included a section highlighting the differential potency of smaller Kaempferol structures in potentiating colistin combinatorial treatment compared to the larger structures. Lastly, concerning the phrase "conducted concurrently with this study," we would like to clarify that both studies were indeed carried out simultaneously. The term "conducted concurrently with this study", therefore, accurately represents the chronology of our research and the study mentioned.

REVIEWER #3 (Remarks to the Author):

The authors have fully addressed my concerns.

We thank this reviewer for their previous suggestions, which were indeed helpful for improving our manuscript.